# Evolutionary alterations in gene expression and enzymatic activities of gibberellin 3-oxidase 1 in *Oryza*

Kyosuke Kawai [1], Sayaka Takehara[1], Toru Kashio[1], Minami Morii[1], Akihiko Sugihara[1], Hisako Yoshimura[1], Aya Ito[1], Masako Hattori[1], Yosuke Toda [2,3], Mikiko Kojima[4], Yumiko Takebayashi[4], Hiroyasu Furuumi[5], Ken-ichi Nonomura [6], Bunzo Mikami[7], Takashi Akagi [8], Hitoshi Sakakibara [4,9], Hidemi Kitano[1], Makoto Matsuoka[1] & Miyako Ueguchi-Tanaka [1✉]

Proper anther and pollen development are important for plant reproduction. The plant hormone gibberellin is important for anther development in rice, but its gametophytic functions remain largely unknown. Here, we report the functional and evolutionary analyses of rice gibberellin 3-oxidase 1 (OsGA3ox1), a gibberellin synthetic enzyme specifically expressed in the late developmental stages of anthers. Enzymatic and X-ray crystallography analyses reveal that OsGA3ox1 has a higher $GA_7$ synthesis ratio than OsGA3ox2. In addition, we generate an *osga3ox1* knockout mutant by genome editing and demonstrate the bioactive gibberellic acid synthesis by the OsGA3ox1 action during starch accumulation in pollen via invertase regulation. Furthermore, we analyze the evolution of *Oryza GA3ox1s* and reveal that their enzyme activity and gene expression have evolved in a way that is characteristic of the *Oryza* genus and contribute to their male reproduction ability.

[1] Bioscience and Biotechnology Center, Nagoya University, Chikusa, Nagoya 464-8601, Japan. [2] Japan Science and Technology Agency, Kawaguchi, Saitama 332-0012, Japan. [3] Institute of Transformative Bio-Molecules, Nagoya University, Chikusa, Nagoya 464-8602, Japan. [4] RIKEN Center for Sustainable Resource Science, Tsurumi, Yokohama 230-0045, Japan. [5] Technical Section, National Institute of Genetics, Mishima, Shizuoka 411-8540, Japan. [6] Plant Cytogenetics, National Institute of Genetics, Mishima, Shizuoka 411-8540, Japan. [7] Division of Applied Life Sciences, The Graduate School of Agriculture, Kyoto University, Uji 611-0011, Japan. [8] Graduate School of Environmental and Life Science, Okayama University, Okayama 700-8530, Japan. [9] Graduate School of Bioagricultural Sciences, Nagoya University, Nagoya 464-8601, Japan. ✉email: mueguchi@nuagr1.agr.nagoya-u.ac.jp

Gibberellins (GAs) are plant hormones with a tetracyclic diterpenoid structure involved in various important developmental processes, including those of floral organs and shoots. GAs are synthesized in a stepwise manner by multiple oxidases, and the conversion from direct precursors to bioactive GAs is mediated by GA 3-oxidases (GA3oxs). Bioactive GAs bind to the GA receptor GIBBERELLIN INSENSITIVE DWARF1 (GID1) and induce a downstream GA response[1]. The synthesized bioactive GAs are inactivated via oxidation by GA 2-oxidases (OsGA2oxs) to regulate their endogenous levels in the plant body[2,3]. One of the rice GA3ox, OsGA3ox2, undergoes negative expression feedback regulation depending on the endogenous level of bioactive GA, whereas OsGA2oxs undergo positive expression feedback regulation[2–5]. Recently, Takehara et al. revealed that OsGA2oxs gradually form multimer structures with increasing GA concentration, which elevates the enzymatic activities through allosteric mechanism[6]. In this way, rice has elaborate systems to regulate the endogenous levels of bioactive GAs.

Several studies on the relationship between anther development and GAs have been conducted to date. Aya et al. showed that GAs regulate the formation of exine, a component of the pollen wall, and programmed cell death in the tapetum cells of the inner wall of the anther, which are involved in anther development[7]. They also revealed that the direct activation of a cytochrome P450 hydroxylase CYP703A3 by GAMYB is key to exine formation[7]. In addition to such sporophytic functions in anthers, GAs are involved in gametophytic functions, such as pollen tube germination and elongation[8]; however, little is known about their precise roles in the gametophytic processes.

In addition, GAs promote shoot elongation in vegetative tissues. Since it is important to regulate plant height for proper growth, rice has sophisticated mechanisms of GA regulation, such as feedback regulation of gene expression of inactivating enzymes and allosteric GA regulation as described above. If these mechanisms do not work properly, and GA accumulates in excess, rice does not grow normally. One example is the "Bakanae disease" of rice, which is caused by the fungal plant pathogen *Gibberella fujikuroi* and is a serious agronomical problem throughout Asia[9]. This fungus has multiple enzyme clusters in its genome and synthesizes large amounts of $GA_3$, one of the bioactive GAs, in the plant body[10]. $GA_3$ is not structurally inactivated by GA2oxs in plants owing to its structural property of having a double bond between the C1 and C2 positions of the *ent*-kaurene skeleton[11]. Another abnormality caused by excessive GA signaling is the *slender rice1* (*slr1*), a loss of function mutant of the *SLR1* gene, which encodes the DELLA protein, a GA signal repressor in rice. This plant shows excessive shoot growth and withering, as if the GA response is constantly active[12].

Despite the existence of such important systems regulating endogenous GA levels in plants, rice anthers contain more than 100 times higher concentrations of bioactive GAs than do leaves[13], far exceeding the need for GA responses by the GID1-GA-DELLA system[1]. However, it is unclear why such excessive endogenous GA is present in anthers and what enzymes and downstream factors are responsible for such excessive amounts.

GA3ox and GA2ox, members of the 2-oxoglutarate (2OG)- and Fe (II)-dependent dioxygenase (2ODD) family, catalyze the oxidation of substrate GAs using 2OG and molecular oxygen as co-substrates and ferrous Fe(II) cofactors[14,15]. The crystal structure of OsGA2ox3 has been reported by Takehara et al., and the residues important for its activity have been clarified[6]. There are nine functional GA2oxs paralogs in rice, and their evolutionary origin has also been reported[16]. Conversely, the crystal structures of GA3oxs in rice have not been elucidated, and the functional differences and evolutionary origin of the two paralogs in rice are also unknown.

In this study, we report the function and evolution of OsGA3ox1, one of the two OsGA3oxs in rice, and address the synthesis of large amounts of bioactive GAs in anthers. From structural and enzymatic analyses, we found that OsGA3ox1 can generate $GA_7$ as well as $GA_4$ from the precursor $GA_9$ at a higher $GA_7$ production rate than that of OsGA3ox2. $GA_7$ is structurally similar to $GA_3$, is not inactivated by GA2oxs[17], and has the highest affinity for the GID1 receptor, making it the most potent bioactive of all forms of GAs. In addition, we show that *OsGA3ox1* is expressed only in the later stage of pollen and lodicule development and produces bioactive GAs in pollen, functioning in a gametophytic manner, such as pollen starch accumulation, pollen tube germination, and pollen tube elongation, to increase the reproductive ability. By analyzing *GA3ox1* genes of *Oryza* plants, we found that the characteristics of expression and enzyme activity of *OsGA3ox1* have evolved specifically in the genus *Oryza*. Based on these analyses, we suggest that the evolution of the enzymatic function and gene expression of *Oryza GA3ox1* contribute to the reproductive advantage of *Oryza* plants.

## Results and discussion

**OsGA3ox1 synthesizes bioactive $GA_7$.** First, we determined the endogenous GA levels in rice (cultivar Nipponbare) anthers. The mature anthers in Nipponbare possessed high levels of $GA_4$ (nearly 400 pmol g$^{-1}$ fresh weight [FW]), which is a bioactive form of GA, whereas relatively less amounts of $GA_1$ (approximately 10 pmol g$^{-1}$ FW) were detected (Fig. 1a). $GA_4$ and $GA_1$ are both bioactive forms of GA, which lack and possess 13-OH, respectively (Supplementary Fig. 1a, b). These results corroborate previous findings, which indicate that bioactive GAs are produced mainly through the non-13-hydroxylation pathway in rice reproductive organs, whereas the 13-hydroxylation pathway occurs in vegetative organs; also that the abundance of bioactive GAs in anthers is higher than that in leaves[18]. In addition to $GA_4$, a substantial amount of $GA_7$ (another bioactive GA without 13-OH), was observed (approximately 300 pmol g$^{-1}$ FW) (Fig. 1a and Supplementary Fig. 1b). Similar to $GA_3$ produced by the "Bakanae" fungus *G. fujikuroi*, $GA_7$ cannot be inactivated by GA2oxs as it possesses a double bond between the C1 and C2 positions of $GA_4$. We believe that $GA_7$ may have the highest efficiency among bioactive GAs in promoting shoot elongation (Supplementary Fig. 1c–e). Using yeast, the GID1-GA-DELLA interaction for all bioactive GAs was explored (Supplementary Fig. 1f). We deduced that $GA_7$ showed the highest efficiency among all GA ligands, followed by $GA_4$.

Next, we examined the causal enzyme for the accumulation of $GA_4$ and $GA_7$ in rice anthers. Rice harbors two *GA3ox* genes[5], which encode GA3-oxidases that catalyze the final step of GA biosynthesis, i.e., the addition of a hydroxyl group at C3 of GA precursors (Supplementary Fig. 1a). *OsGA3ox2* is expressed in various organs, such as shoots and roots[5] (Supplementary Fig. 2a), and its null mutant alleles exhibit a severe dwarf phenotype[19]. *OsGA3ox1* is expressed specifically in anthers (Supplementary Fig. 2b). *OsGA3ox1* promoter-GUS analysis revealed expression in the late developmental stage of pollen, anther filaments, and lodicules, but not in vegetative organs (Supplementary Fig. 2c, d). To examine the role of *OsGA3ox1 in planta*, we generated *osga3ox1* knockout mutant plants using the CRISPR/Cas9 system (Supplementary Fig. 3a–e) and determined endogenous GA levels in the anthers (Fig. 1a). Endogenous $GA_4$ and $GA_7$ levels were significantly decreased, whereas the abundance of precursor $GA_9$ increased in *osga3ox1*. This suggests that OsGA3ox1 is the major enzyme catalyzing the synthesis of $GA_4$ and $GA_7$ in rice anthers. Next, we determined the GA biosynthesis activities of

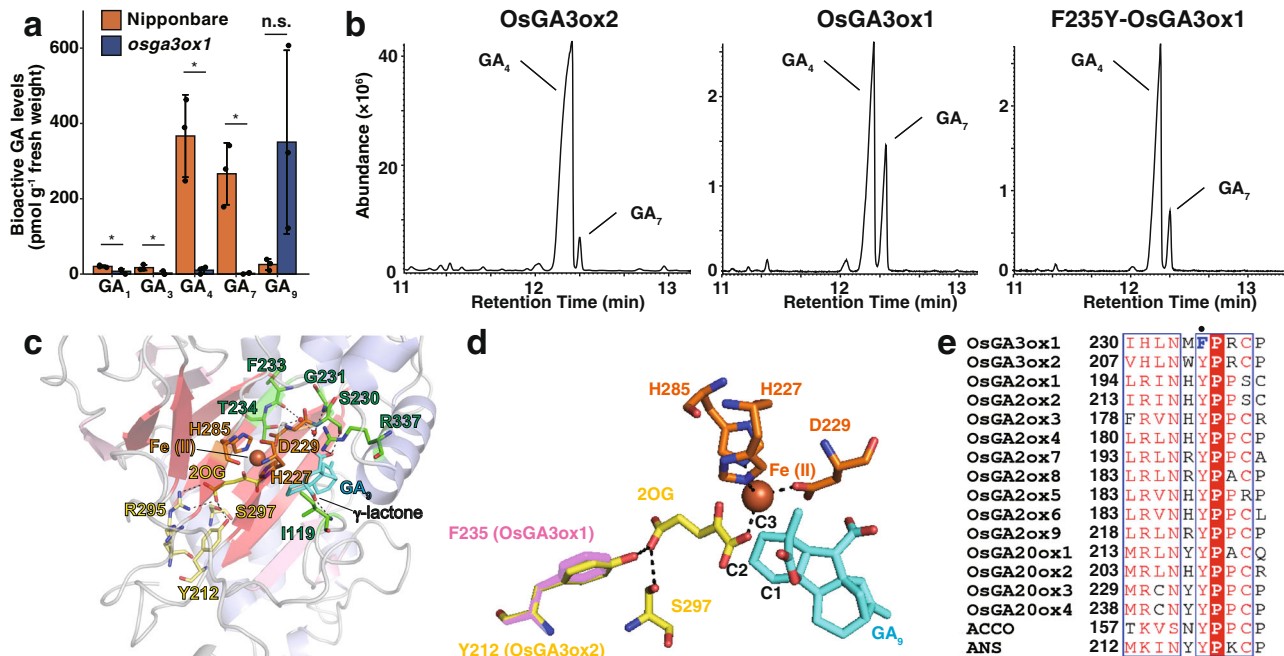

**Fig. 1 Activity of rice GA3ox1 and GA3ox2 upon synthetic gibberellic acid, and their structural factors. a** Endogenous GA levels in anthers of wild-type (cultivar Nipponbare) and *osga3ox1* mutant. Error bars, s.d. $n = 3$ Student's *t* test (\*\**p* < 0.05). **b** Gas chromatogram of the reaction mixture for OsGA3ox2 (left panel), OsGA3ox1 (middle panel), and the OsGA3ox1-F235Y mutant protein (right panel) as a substrate $GA_9$. The positions of the products $GA_4$ and $GA_7$ are indicated. **c** Structures of OsGA3ox2 around the GA-binding site with $GA_9$ and the co-substrates, Fe (II), and 2OG (see also Supplementary Figs. 4–6). Fe (II) is represented as an orange sphere and 2OG and $GA_9$ are indicated in yellow and blue, respectively. The amino acids establishing interactions with Fe (II) (H227, D229, and H285), 2OG (Y212, R295, and S297), and $GA_9$ (I119, S230, G231, F233, T234, and R337) are shown in orange, yellow, and green, respectively. **d** Structural comparison of the active sites of OsGA3ox2 and OsGA3ox1 (predicted structure). Important amino acids are shown with the same colors and numbers as in (**c**) (OsGA3ox2). The site with amino acid Y212 or F235 in the 2OG-interacting site is responsible for the substitution in OsGA3ox1 (shown in **e**). **e** Sequence alignment of rice GA oxidase enzymes, OsGA3oxs, OsGA20oxs, OsGA2oxs, and other 2ODD family proteins, ACCO (ACC oxidase), and ANS (anthocyanidin synthase). The black dot indicates F212 in the 2OG-interacting site of OsGA3ox2 as described for (**c**) and (**d**).

recombinant OsGA3ox1 and OsGA3ox2 for $GA_9$ as a substrate in vitro. Whereas the main product of OsGA3ox2 was $GA_4$ (Fig. 1b; OsGA3ox2), we found a remarkable peak of $GA_7$ as well as $GA_4$ when we analyzed OsGA3ox1 (Fig. 1b; OsGA3ox1). This result indicates OsGA3ox1 produces $GA_7$ in a higher $GA_7$ synthesis ratio than that of OsGA3ox2.

**X-ray crystal analysis of OsGA3ox2.** To understand why OsGA3ox1 produced higher levels of $GA_7$ compared to OsGA3ox2, we performed X-ray crystallography analyses of these enzymes. As a result, we succeeded in obtaining the structure of the OsGA3ox2 complex with (co-) substrates; $GA_9$ and 2-oxoglutarate (2OG) at 1.9 Å resolution (Fig. 1c and d, Supplementary Figs. 4, 5, and Supplementary Table 1). Fe (II) was eliminated for crystallization to prevent the enzyme reaction, as previously described[6]. The carboxyl group of $GA_9$ established interaction with S230, G231, F233, T234, and R337 directly or indirectly through water, and its γ-lactone ring established interaction with the main chain of I119 (Fig. 1c, Supplementary Fig. 5a). These amino acids are conserved between OsGA3ox1 and OsGA3ox2, but not between OsGA2oxs, OsGA20oxs, or even AtGA3oxs (Supplementary Fig. 6), with the exception that I119 is replaced with Y in OsGA3ox1. These residues of OsGA2ox3, whose structures were resolved in our previous study[6], were not associated with GA-binding (Supplementary Figs. 5b, c, 6) despite conservation between OsGA3ox2 and OsGA2ox3 (e.g., I119 in OsGA3ox2 for γ-lactone ring interaction and the corresponding residue, I98 in OsGA2ox3 for no interaction with GA). Moreover, upon comparing the structure of OsGA3ox2 with that of OsGA2ox3, we found that the GAs were inverted in the substrate-

binding pocket. These data suggest that GA-related 2ODDs acquired various GA-binding features during enzyme evolution. Conversely, the residues H227, D229, and H285 of OsGA3ox2 established interaction with Fe (II); Y212, S297, and R295 established interaction with 2OG (Fig. 1c and Supplementary Fig. 6). These amino acids are highly conserved in almost all 2ODDs, except F235 in OsGA3ox1, which is a conserved Y residue in other 2ODDs (Fig. 1c–e and Supplementary Fig. 6)[20,21]. Hence, we speculated that this substituted residue could be responsible for $GA_7$ hyperproduction. To confirm this, we produced the mutated OsGA3ox1 protein having the F235Y like other 2ODDs and measured its enzyme activity (Fig. 1b; F235Y-OsGA3ox1). The productivity of $GA_7$ was lower in F235Y-OsGA3ox1 than that in OsGA3ox1. Taken together, these results indicate that the substitution of a highly conserved Y with F in OsGA3ox1 contributes to the hyperproductivity of $GA_7$.

***osga3ox1* mutant shows gametophytic male sterility.** The *osga3ox1* mutant did not exhibit any abnormal phenotype during the vegetative stages (Supplementary Fig. 7a, b). However, compared with the wild type, in the reproductive stages, the *osga3ox1* mutant presented with non-elongated anther filaments in the flowering stage, abnormal anther dehiscence, and no spikelet opening; but it showed normal pistils (Supplementary Fig. 7c–f and Supplementary Videos 1 and 2). The *osga3ox1* showed abnormal starch accumulation in mature pollen grains (Fig. 2a).

Similar phenotype was observed using two independent mutant alleles, *osga3ox1-cr1/osga3ox1-cr1* and *osga3ox1-cr7/osga3ox1-cr7* (Supplementary Fig. 3e). We mainly used *osga3ox1-cr1/osga3ox1-cr1* for further analyses as *osga3ox1*. Upon $I_2$-KI staining, 81% of

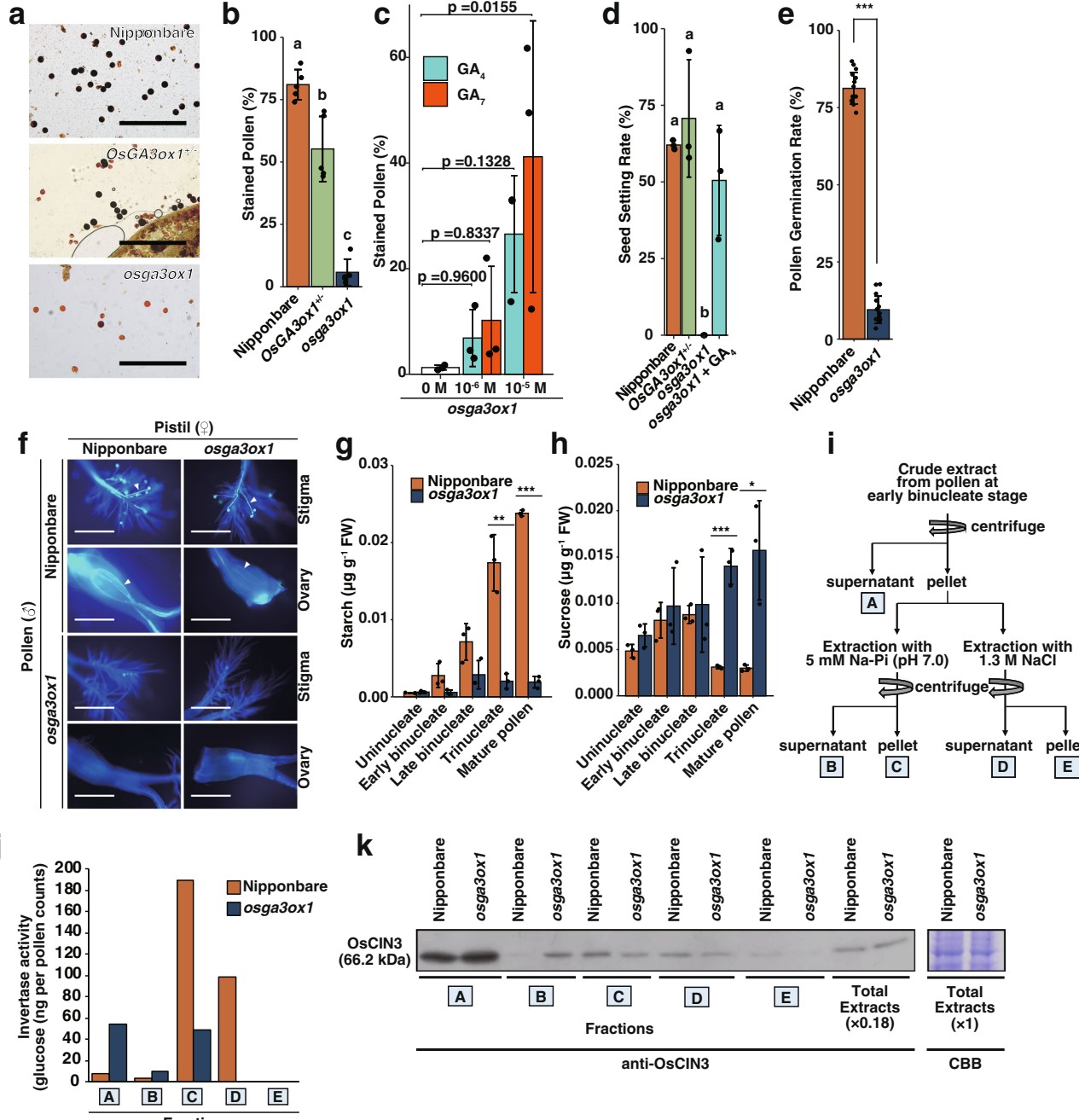

**Fig. 2 Phenotypic analysis of the reproductive organs of *osga3ox1* mutant rice. a**, **b** Pollen viability test of Nipponbare, *OsGA3ox1*$^{+/-}$ (heterozygous), and *osga3ox1* mutant performed by conducting staining with the $I_2$-KI solution. Pollen grains stained black are deemed viable; those stained red are deemed sterile (Bars = 0.5 mm) (**a**). Percentage of viable vs sterile ($n = 5$) (**b**). **c** Pollen viability test of GA-treated *osga3ox1* mutant performed by conducting $I_2$-KI staining, same as that for panel (**b**) ($n = 3$). *P*-values by Dunnett's multiple comparison test are indicated. **d** Seed setting rate of Nipponbare, *OsGA3ox1*$^{+/-}$, *osga3ox1*, and $GA_4$-treated *osga3ox1* mutant plants ($n = 3$). **e** In vitro pollen tube germination of Nipponbare and *osga3ox1* mutant ($n = 15$). **f** Reciprocal crossing test between Nipponbare and *osga3ox1* mutant plants. Arrowheads indicate pollen tubes. Bars = 0.5 mm. **g**, **h** Endogenous quantities of starch (**g**) and sucrose (**h**) contents in anthers of Nipponbare and *osga3ox1* mutant ($n = 3$). **i**, **j** Enzyme activity of acid invertase in extractions from Nipponbare and *osga3ox1* mutant pollen in the early binucleate stage. Summary of the extracted fractions (**i**) and their invertase activities (**j**) (these experiments were repeated three times with similar results). **k** Western blot analysis of OsCIN3 in extracted fractions in (**l**) and total extract of anthers using an OsCIN3 specific antibody. CBB staining of total extract is also shown. In all experiments, error bars = s.d. For panels (**b**) and (**d**): one-way ANOVA with Tukey's multiple comparisons test. Different letters denote significant differences ($p < 0.001$). For panels (**e**), (**g**), and (**h**): ***$p < 0.001$, **$p < 0.01$, and *$p < 0.05$ by two-tailed paired *t*-tests.

the Nipponbare (wild-type) pollen grains were stained, whereas only 55% of the *OsGA3ox1*$^{+/-}$ (heterozygous) and 4% of the *osga3ox1* mutant (homozygous) pollen grains were stained (Fig. 2a, b, Supplementary Fig. 3e). The abnormality of the *osga3ox1* was

restored by exogenous GA application; notably, $GA_7$ recovered at a lower concentration than did $GA_4$ (Fig. 2c). *osga3ox1* showed a complete loss of seed setting ability (Fig. 2d). In addition, the pollen grains of *osga3ox1* did not germinate or elongate a tube,

whereas its pistil functioned normally (Fig. 2e and f, Supplementary Fig. 7g, h). These data suggest that *osga3ox1* exhibits gametophytic male sterility. The mutant phenotypes were consistent with their tissue-specificities as shown in the promoter-GUS analysis (Supplementary Fig. 2c, d).

Fluorescein diacetate (FDA) and 4′,6-diamidino-2-phenylindole (DAPI) staining revealed that the *osga3ox1* mutant pollen shows normal cell development (Supplementary Fig. 7i). The analysis of starch and sucrose in anthers revealed that starch synthesis, in particular the conversion step of glucose from sucrose that is mainly regulated by invertases[22], was defective in *osga3ox1* mutant pollen (Fig. 2g, h). Rice has nine *cell wall invertases* (*CIN*s), eight *neutral invertases* (*NIN*s), and two *vacuolar invertases* (*VIN*s)[23]. We speculated that a cell wall invertase, *OsCIN3*, is likely one of the targets of bioactive GAs synthesized by OsGA3ox1, as its expression shows appreciable correlation with that of *OsGA3ox1* in anthers (Supplementary Figs. 2b, 7j, 8)[23,24]. The robust induction of *OsCIN3* was observed upon exogenous $GA_4$ treatment in both wild-type and *osga3ox1* mutant (Supplementary Fig. 7k). Despite the induction of *OsCIN3* upon GA application, downregulation of *OsCIN3* expression was not observed in the *osga3ox1* mutant. To examine the detailed relationship between GAs produced by OsGA3ox1 and invertase activities in early maturated pollen, we analyzed the in situ invertase activity (Supplementary Fig. 7l) and enzyme activities of fractionated pollen proteins (Fig. 2i, j). These results indicate that the activities of invertase in fractions D and E, which are associated with the cell wall, were decreased in the *osga3ox1* mutants. In addition, we performed a western blot analysis on the anther extract fractions with an OsCIN3 specific antibody (Fig. 2k). These results indicate that in Nipponbare extracts, only a small amount of OsCIN3 protein could be detected in fraction B, which was extracted with sodium phosphate buffer, whereas a large amount of OsCIN3 was detected in same fraction in *osga3ox1* extracts. Conversely, a large amount of OsCIN3 was detected in Nipponbare fraction C and D, which was strongly bound to the cell wall. This result indicates that, at least, OsCIN3 is an invertase—properly retained in the cell wall in Nipponbare —with reduced activity due to incorrect retention in the cell wall in *osga3ox1*. Further research is needed to clarify the mechanism by which GA is involved in proper retaining of OsCIN3. Together, we concluded that a high abundance of $GA_4$ and $GA_7$ produced by OsGA3ox1 in pollen played an important role in gametophytic reproduction through starch synthesis, probably via the regulation of OsCIN3.

Previously, Aya et al. reported that various rice GA-related mutants, such as *ent-copanyl diphosphate synthase* (*cps*), *gid1*, and *gamyb*, showed defects in programmed cell death of tapetal cells and the formation of exine and Ubisch bodies, resulting in pollen disruption in a sporophytic manner[7]. We consider that GAs function in various developmental stages of anther and pollen, and the causal genes of the mutants previously reported[7] were expressed from the early stage of anther development. As these mutants exhibit the disruption of pollen in the early anther developmental stage through their sporophytic defections, researchers could not analyze the gametophytic GA functions in male organ development in the later stage. In contrast, *OsGA3ox1* is a good candidate for the analysis of gametophytic functions of GA at the later stage of anther development[25] (Supplementary Fig. 2c).

**Evolution of GA$_7$ productivity of *Oryza* GA3ox1.** As OsGA3ox1 can produce substantial amounts of $GA_7$ compared to OsGA3ox2, we speculated that OsGA3ox1 gained a new function in male fertility through the substitution of Y to F in the 2OG-interacting site, that is, $GA_7$ producing activity of OsGA3ox1. Therefore, we examined the mechanism and juncture of acquiring this substitution during enzyme evolution. The phylogenetic analysis of *GA3ox* orthologs derived from land plants (Supplementary Table 2) indicated that the rice *GA3ox* orthologous genes underwent duplication after Poaceae emerged (Supplementary Fig. 9a), one is the ancestor of *OsGA3ox1* (hereafter referred to as "Poaceae *GA3ox1* orthologs"), and the other is *OsGA3ox2* (hereafter referred to as "Poaceae *GA3ox2* orthologs"). Poaceae *GA3ox1* orthologs have been lost in certain Poaceae species, such as barley and wheat, whereas Poaceae *GA3ox2* orthologs exist in all Poaceae species examined (Supplementary Fig. 9a, b), as previously reported[26]. We investigated whether the substitution of Y to F in the 2OG-binding site occurred before the genus *Oryza* emerged; however, none of the Poaceae GA3oxs orthologs exhibited amino acid substitutions (Supplementary Fig. 9c) except for rice. Therefore, we focused on the *GA3ox1* orthologs in the genus *Oryza*.

We next combined public whole-genome data (Supplementary Table 2) and direct sequencing data of *GA3ox1*s obtained herein using genomic DNA of 19 *Oryza* species obtained from Wild Core Collection in the National Institute of Genetics. GA3ox1 orthologs from FF to AA genome species[27–29] demonstrate F at the corresponding residue (referred to as F-type) (Fig. 3a). Amino acid residues for 2OG and Fe (II) binding are highly conserved, except F235. On the other hand, the GA3ox1 orthologs of GG, HHJJ genome species, and *L. perrieri*, possess Y (referred to as Y-type). These results indicated that the substitution occurred in the ancestor of FF to AA genomes (Fig. 3b). We also isolated full-length cDNA of GA3ox1 orthologs from six *Oryza* plants and *L. perrieri* and determined the productive ratio of $GA_7/GA_4$ in vitro (Fig. 3c). The ratio of GA3ox1 in *O. granulata* (GG) was <0.1, whereas that of *O. brachyantha* (FF) was >0.2. Notably, the ratio increased during evolution from the EE to AA genome, with >0.5 in *O. sativa*. For *L. perrieri*, the ratio was about 0.3. We found that the amino acid residues that bind to the substrate $GA_9$, 2OG, and Fe(II) were conserved in GA3ox1 of *Oryza* genus and *L. perrieri* except for F235 (Supplementary Fig. 10a). Using structural prediction and the machine learning method of AlphaFold2[30], we investigated the cause of the high $GA_7$ production rate of *L. perrieri* GA3ox1. The structural prediction suggested that Arg315 (corresponding to Arg318 in OsGA3ox1) at the interaction site with 2OG was oriented differently only in *L. perrieri* GA3ox1, resulting in reduced hydrogen bonding (Supplementary Fig. 10b-e). Therefore, the $GA_7/GA_4$ ratio of *L. perrieri* GA3ox1 may also be because its binding to 2OG differs from that of OsGA3ox2.

**Expression patterns of *Oryza* GA3ox1.** Expression data of the Poaceae species, *Triticum aestivum*, *Hordeum vulgare*, *Brachypodium distachyon*, *Sorghum bicolor*, and *Zea mays* revealed that several GA3oxs homologs were expressed in anthers but did not exhibit anther-specific characteristics as *OsGA3ox1* (Supplementary Fig. 11). Next, we compared the tissue specificity and intensity of *GA3ox1* orthologs among six *Oryza* species and *L. perrieri* by performing quantitative reverse-transcription polymerase chain reaction (qRT-PCR) (Fig. 4a–g). *GA3ox1* orthologs of the *Oryza* AA genome species (*O. sativa* and *O. rufipogon*) showed strict anther specificity and high expression levels (Fig. 4a, b). The *GA3ox1* orthologs of *O. punctata* (BB) and *O. australiensis* (EE) were expressed in anthers, roots, and whole spikelets (Fig. 4c, d). In *O. brachyantha* (FF) and *O. granulata* (GG), *GA3ox1* orthologs were also detected in roots or other organs (Fig. 4e, f), and the expression levels in anthers were lower than those in the anthers of AA genome species. In *L. perrieri*, the expression showed anther specificity, but at extremely low levels

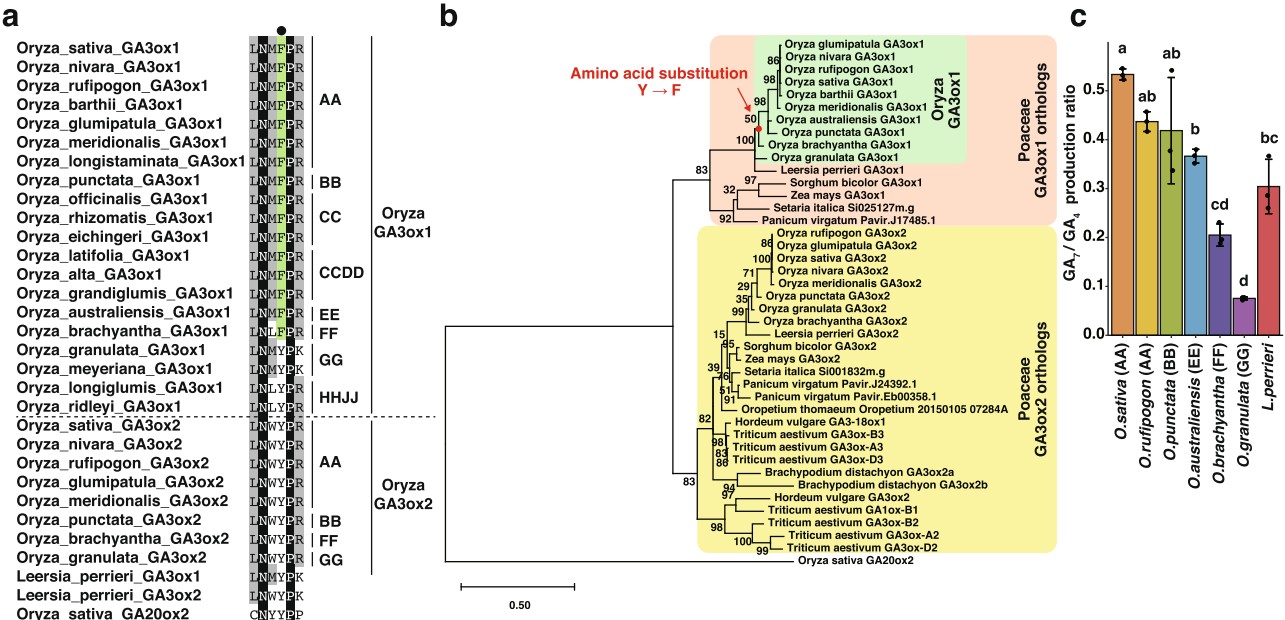

**Fig. 3 Residues in the 2OG-interacting site of Poaceae GA3ox orthologs and enzymatic activity of *Oryza* GA3ox1. a** Amino acid sequence alignment in the 2OG-interacting site around F (in OsGA3ox1) of Poaceae GA3ox orthologs and OsGA20ox2. Residues of F in the interacting site of 2OG are indicated in green, those that are 100% identical are shown in black, and those with more than 50% identity are depicted in gray. The black dot indicates the amino acid in the interacting site with co-substrate 2OG, F (shown in green), or Y. On the right side of the alignment, the letters AA and BB indicate the AA and BB *Oryza* genomes, respectively. **b** Phylogenetic tree of Poaceae GA3ox orthologs and *OsGA20ox2*. The red dot indicates the common ancestor of *O. sativa* to *O. brachyantha* that acquired the residue F in the active site. **c** In vitro enzyme activity of GA3ox1 of *Oryza* species and *L. perrieri*; *O. sativa* GA3ox1, *O. rufipogon* GA3ox1, *O. punctata* GA3ox1, *O. brachyantha* GA3ox1, *O. granulata* GA3ox1, *L. perrieri* GA3ox1. Error bars, s.d. $n = 3$, one-way ANOVA with Tukey's multiple comparisons test. Different letters denote significant differences ($p < 0.05$).

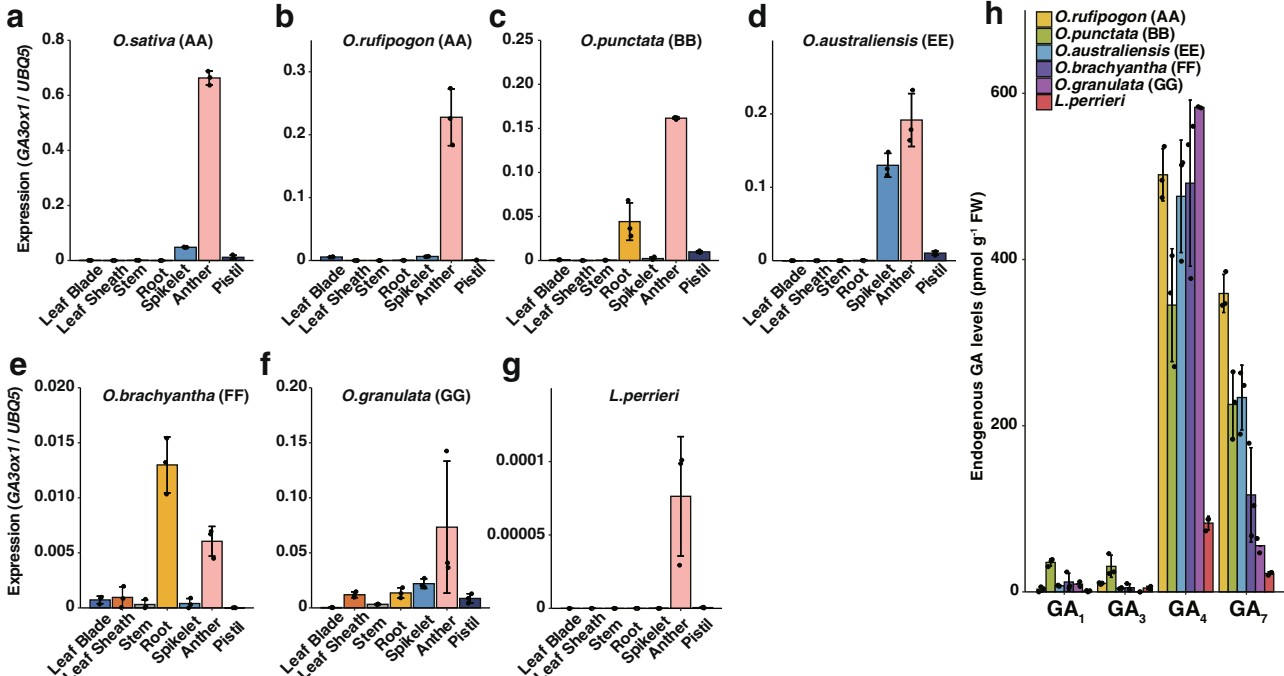

**Fig. 4 Expression analysis of *GA3ox1* and endogenous gibberellic acid levels in *Oryza* species and *L. perrieri*. a–g** Gene expression analysis of *GA3ox1* orthologous genes in *Oryza* species and *L. perrieri*; *O. sativa* (**a**), *O. rufipogon* (**b**), *O. punctata* (**c**), *O. australiensis* (**d**), *O. brachyantha*, (**e**), *O. granulata* (**f**), *L. perrieri* (**g**) in leaf blades, leaf sheaths, stems, roots, spikelets, anther, and pistils. Error bars, s.d. n = 3 for each sample. **h** Endogenous GA levels in anthers of *Oryza* species and *L. perrieri*. Error bars, s.d. $n = 3$ (*O. rufipogon, O. punctata, O. australiensis, O. brachyantha*, and *L. perrieri*) or $n = 2$ (*O. granulata*).

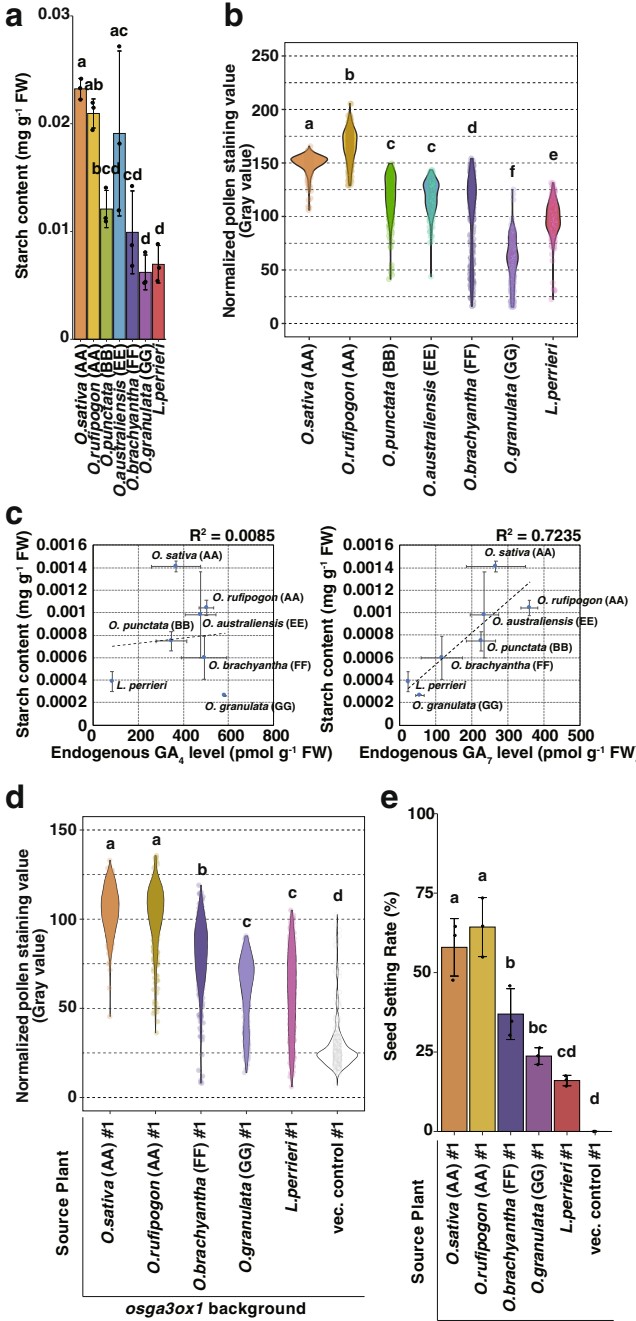

**Fig. 5 Effects of *GA3ox1* on pollen and seed fertility in *Oryza* species and *L. perrieri*. a** Endogenous quantities of starch in *Oryza* species and *L. perrieri* anthers. Error bars, s.d. $n = 3$, one-way ANOVA with Tukey's multiple comparisons test. Different letters denote significant differences ($p < 0.05$). **b** Quantification of the staining value of KI-stained *Oryza* species and *L. perrieri* pollen. $n = 300$ pollen grains for each species, statistical analysis as mentioned in panel (**a**). **c** Correlation between starch content and endogenous gibberellin levels in *Oryza* species and *L. perrieri* anthers. The coefficient of determination ($R^2$) is shown on the upper right side of the plot. The respective values are 0.0085 between starch content and $GA_4$ levels (left panel), and 0.7235 between starch content and $GA_7$ levels (right panel). **d** Quantification of the staining value of KI-stained pollen grains of transgenic plants described in Supplementary Fig. 12a and vector control. $n = 300$ pollen grains for each line; statistical analysis as mentioned in panel (**a**). **e** Seed setting rate of transgenic plants described in Supplementary Fig. 12a and vector control. $n = 3$ panicles for each line; statistical analysis as mentioned in panel (**a**).

Pollen starch accumulation is important for pollen tube germination and elongation and contributes to male reproductivity. Previously, Lee et al. [31] compared the pollination of *adp-glucose pyrophosphorylase* (*osagpl4*) mutant, defective in starch synthesis, with that of homozygous ($OsAGPL4^{-/-}$) and heterozygous ($OsAGPL4^{+/-}$) mutant anthers. They showed that 10–20% of $OsAGPL4^{-/-}$ inheritors could be detected in the descendants of $OsAGPL4^{-/-}$ plants, whereas none were detected in $OsAGPL4^{+/-}$ plants, suggesting that the pollen grains with high starch accumulation out-competed those with low starch. Analogously, increasing starch content in pollen during the evolution of *Oryza GA3ox1* orthologs might have conferred reproductive advantages. We found a correlation between starch content and endogenous $GA_7$ in pollen; however, no correlation between starch and $GA_4$ was observed (Fig. 5c), suggesting that endogenous $GA_7$ is important for starch accumulation in pollen.

To compare the effects of both transcription and enzyme activity on the reproductive phenotypes in the same background, we conducted a complementation test by introducing the genomic region of *GA3ox1* orthologs of five *Oryza* and *L. perrieri* plants into the *osga3ox1* mutant background (Supplementary Figure 12a). $K_2$-KI staining and seed setting rate analyses revealed an appreciable correlation with $GA_7$ productivities of GA3ox1 orthologs (Figs. 4h, 5d, e, Supplementary Fig. 12b, c). The transformants of *L. perrieri GA3ox1* showed low expression levels of *GA3ox1* in the anthers, while the transformants of *O. granulata GA3ox1* showed slightly higher expression levels of *GA3ox1* in the leaf blades (Supplementary Fig. 12d). Consistent with each expression level and enzymatic activity, no significant difference was detected in the vegetative phenotype (Supplementary Fig. 12e, f). We also measured the endogenous levels of bioactive GAs in the anthers of all transgenic lines (Supplementary Fig. 12g). The amount of endogenous $GA_7$ in the anthers of the lines that possess *GA3ox1* of AA genome plants (*O. sativa* and *O. rufipogon*) was higher than that in the anthers of other lines, consistent with their reproductive phenotypes. These results indicate that the ability to synthesize $GA_7$ as well as $GA_4$ may be important, and is likely related to pollen starch accumulation and improved reproductive performance.

Here, we report the analysis of the enzyme and expression evolution of GA3ox1 in wild rice after the establishment of the *Oryza* species. The Y to F amino acid substitution in the 2OG interaction site accounts for $GA_7$ production in the AA-FF genome species. In line with the evolution of enzymatic activities, gene expression has been localized mainly in the male organ, but not in the shoot where excessive bioactive GAs can promote

(approximately ten-thousandth of that of *OsGA3ox1*) (Fig. 4g). These results indicated that the expression pattern of *GA3ox1* underwent evolution to gain anther-specific features and underwent enhancements during genus *Oryza* evolution. The endogenous $GA_7$ levels were consistent with the enzyme activities and the expression analysis data (Fig. 4h).

**$GA_7$ is important for the fertility of *Oryza*.** As the pollen grains of *osga3ox1* mutant showed abnormality in starch accumulation (Fig. 2a, b, g), we analyzed starch content in anthers obtained from six *Oryza* and *L. perrieri* species (Fig. 5a). The further the phylogenetic distance to *O. sativa*, the lower the levels of endogenous starch in anthers. Similar to the endogenous starch levels, the performance of $I_2$-KI staining in pollen revealed that AA genome species accumulated more starch than other species (Fig. 5b).

harmful overgrowth. These enzymatic and expression evolutions might have made it possible for rice to contain high endogenous levels of bioactive GAs in an anther-specific manner. These large amounts of GAs contribute to the accumulation of starch through the regulation of invertase (Fig. 2g, h, j, k). Given the male sterility phenotype of *osga3ox1*, during evolutionary processes, *GA3ox1* may have become crucial for male fertility in rice. This unique GA$_7$ synthesis strategy in *Oryza* might confer certain reproductive advantages for its successful global dissemination[32]. Altogether, this study paves the way for a better understanding of the factors of dissemination and the reproductive systems of *Oryza*.

## Methods

**Plant material, growth conditions, and genotyping**. Seeds, whole plants, and genomic DNA extracts of wild rice species were obtained from the Resource Bank of the National Institute of Genetics (http://www.shigen.nig.ac.jp/rice/oryzabase/). The accession numbers of wild rice used in this study are: *Oryza rufipogon*, W0106; *Oryza punctata*, W1514; *Oryza officinalis*, W1361; *Oryza rhizomatis*, W1805; *Oryza eichingeri*, W1527; *Oryza latifolia*, W1166; *Oryza alta*, W0017; *Oryza grandiglumis*, W0613; *Oryza australiensis*, W0008; *Oryza brachyantha*, W0654; *Oryza granulata*, W0002; *Oryza meyeriana*, W1356; *Oryza longiglumis*, W1220; *Oryza ridleyi*, W0001; and *Leersia perrieri*, W1529.

*Oryza sativa* L. "Nipponbare", the *osga3ox1* mutant and wild species, seeds were hulled and surface-sterilized with 70% ethanol and 50% sodium hypochlorite. After sterilization, seeds were germinated and subjected to growth conditions on 0.5 × Murashige and Skoog (MS) media containing 0.8% gellan gum (Wako) and 25 mg/l meropenem (Wako) for selection of the *osga3ox1* mutant. After germination under continuous light conditions at 30 °C for 2 weeks, seedlings were transplanted to a pot for growth in a greenhouse at a temperature of 28 °C and 60% humidity under 16 h light (long-day treatment) conditions. After approximately 2 months of subjection to the above-mentioned growth condition, plants were subjected to a 10 h light (short-day treatment) condition to induce reproductive growth.

To perform genotyping for *osga3ox1*, the PCR reactions were conducted using the primers listed in Supplementary Table 3. After the completion of PCR amplification, DNA fragments were subjected to digestion at 37 °C for 2 h with *Kpn*I in 10 μl of the following mixture: 10 units *Kpn*I, 1 μl 10× buffer, and 2 μl of PCR-amplified DNA fragments. Then, the digested DNA fragments were subjected to gel electrophoresis; the presence of a 949 bp fragment indicates the presence of the *osga3ox1* mutant, whereas *Kpn*I digestion (presence of both 519 bp and 429 bp bands) indicates the presence of the wild-type (*OsGA3ox1*$^{+/+}$); heterozygous (*OsGA3ox1*$^{+/−}$) plants were identified based on the presence of all three bands (519 bp, 429 bp, and 949 bp) were present. The four alleles of *osga3ox1* were confirmed by sequencing. The T$_1$ generations of homozygous mutants (*osga3ox1-cr1/osga3ox1-cr1* and *osga3ox1-cr7/osga3ox1-cr7*) and heterozygous mutants (*OsGA3ox1/osga3ox1-cr1* and *OsGA3ox1/osga3ox1-cr7*) were obtained by GA treatment of the T$_0$ generations. We mainly used *osga3ox1-cr1/osga3ox1-cr1* and *OsGA3ox1/osga3ox1-cr1* for the analyses.

**Plasmid construction and rice transformation**. PCR amplification for all constructs was performed using high-fidelity PrimeStar-GXL DNA polymerase (Takara Bio Inc., cat# R050A) or KOD FX Neo (TOYOBO, cat# KFX-201). Primer sequences are listed in Supplementary Table 3.

For constructs of the GST-tagged proteins of OsGA3ox1 and OsGA3ox2, the full-length coding sequences for OsGA3ox1 and OsGA3ox2 were amplified, with the full-length cDNA provided by the National Institute of Agrobiological Sciences (NIAS) used as the PCR template and subcloned into pGEX6P1 (Cytiva). F235Y-OsGA3ox1 was constructed by performing PCR using mutagenized primers with pGEX6P1/OsGA3ox1 as the template.

The *osga3ox1* mutant was generated using the CRISPR/Cas9 system following the method reported by Mikami et al.[33]. To identify a suitable protospacer adjacent motif (PAM), CRISPR-P (http://cbi.hzau.edu.cn/cgi-bin/CRISPR) was used. Concentrations of 2 μM of the oligo-DNA fragments of sequences corresponding to the guide-RNAs listed in Supplementary Table 3 were incubated at 95 °C for 5 min. Then, they were annealed at 25 °C for 20 min. The cloning vector (pU6gRNA) digested with *Bbs*I and oligo-DNA fragments were ligated and transformed into *E. coli* competent cells (XL10-gold). Using competent cells, plasmid DNA extraction was performed based on the alkaline lysis method[34]. The inserted target sequence in the obtained plasmids was sequenced to ensure that no unintentional mutations were introduced. The cloning vectors mentioned above were digested with I-*Sce*I, and the spectinomycin-resistant binary vector pZDgRNA_Cas9ver.2_HPT was digested with *Asc*I and *Pac*I. The digested DNA fragments containing the guide-RNAs sequence and binary vector were ligated and transformed into *E. coli* competent cells (XL10-gold). Plasmid DNA extraction was then performed using competent cells by adopting the alkaline lysis method[34]. Then, to confirm the correct insertion of fragments containing the guide-RNA sequence into a binary vector, DNA was digested by *Bam*HI, and products were visualized by gel electrophoresis.

For promoter-GUS constructs, the 3.3 kb region upstream of the 2.6 kb promoter to the middle of the second exon of *OsGA3ox1* was amplified by PCR using the primers listed in Supplementary Table 3. The amplified DNA fragments and the kanamycin-resistance vector pBI101-Hm3 were digested with *Xba*I and *Sma*I, respectively, and the fragments were inserted into the vector pBI101-Hm3 by ligation. The plasmid DNA was transformed into competent cells (XL10-gold), and extracted using the alkaline lysis method[34]. The inserted DNA fragment sequence was confirmed by sequencing.

For the wild rice *GA3ox1* construct, genomic DNA fragments (*O. sativa*, 7.8 kb; *O. brachyantha*, 5.9 kb; *O. granulata*, 5.6 kb; *L. perrieri*, 5.2 kb) containing the full-length wild rice *GA3ox1* genes were isolated from genomic DNA extract derived from leaves of each plant by PCR using KOD FX Neo (TOYOBO, cat# KFX-201) and primers listed in Supplementary Table 3. These amplified PCR fragments were cloned into the pCR4 Blunt-TOPO vector (Invitrogen, cat# K2875J10) and transformed into *E. coli* competent cells (XL10-gold). Plasmid DNA extraction was performed using competent cells by adopting the alkaline lysis method[34], and successful transformations were confirmed by sequencing. For *GA3ox1* obtained from *O. sativa* and *O. rufipogon*, to prevent it from being targeted by CRISPR/Cas9, the target sequence (5′-CCTCCGGTACCCGAAGCAGATG-3′) was synonymously substituted as below; 5′-GCTCCGGTACCCGAAACAAATG-3′. Using these plasmids as templates, a second PCR was conducted using the PrimeSTAR GXL DNA polymerase (Takara Bio Inc., cat# R050A) and the primers listed in Supplementary Table 3. The resulting PCR fragments contained a *Hind*III linker sequence at the 5′-end and a *Kpn*I linker at the 3′-end; the pSTARA R-5 vectors (Kumiai Chemical Industry Co., Ltd.) were digested with *Kpn*I and *Hind*III. The digested PCR fragments were inserted into the pSTARA R-5 vector using the NEBuilder HiFi DNA Assembly Kit (New England BioLabs) and were transformed into *E. coli* One Shot ccdB Survival 2 T1R competent cells (Invitrogen, cat# A10460). The extracted plasmid DNA was introduced into the *Agrobacterium tumefaciens* strain EHA101 and was used to infect *osga3ox1* mutant rice calli[35]. After subjection to growth continuous of light at 30 °C for 2 weeks, all plants were planted in a pot and grown in a greenhouse.

**GA treatment of rice seedlings**. For the GA treatment response test, sterilized tangibozu seeds[36] were germinated in half-volume Murashige Skoog medium; after 2 days, 1 μL of GA ($10^{-12}$, $10^{-11}$, $10^{-10}$, and $10^{-9}$ M; in 50% acetone) was added to the coleoptile and leaf-sheath junction of each seedling. For the control, 1 μL of 50% acetone was added. Six days later, the length of the second leaf sheath was measured.

For the effects of gibberellin on shoot elongation of Nipponbare, GA$_3$, GA$_4$, and GA$_7$ were diluted with ethanol to prepare a stock solution, which was further diluted to the described concentration and used to treat the rice plants. The control group was treated with the same volume of ethanol as the stock solution diluted in the same manner.

**KI staining of pollen grains and determination of starch content in anthers**. For evaluation of pollen viability, two fresh rice anthers per spikelet were randomly collected in 10 μl of KI solution [1% (v/v) of I$_2$ in 3% (v/v) KI] and were crushed well. Then, 2 μl of the solution was sampled on a slide glass and observed with a stereoscopic microscope using a bright field. Pollen grains that were round and exhibited black coloration after staining were assessed as viable, and yellow- or light red-stained pollen grains were considered sterile.

To evaluate the normalized pollen staining value, white balances of micrographs of KI staining were uniformed and converted to hue, saturation, value (HSV) color space, and then gray values of each pollen were evaluated using the ImageJ script (Supplementary Code 1).

**Evaluation of starch content in pollen grains**. The frozen anther samples were homogenized using 80% ethanol. The insoluble material was collected by centrifugation and was subjected to washing steps using hot 80% ethanol. After centrifugation, the pellet was used for starch analysis. The supernatant was treated with activated charcoal and was filtered to remove ethanol-soluble materials, which could interfere with the assay; then, the sample was used for glucose and sucrose analysis. Sucrose was hydrolyzed into glucose and fructose by invertase (Sigma, cat# I4504), and content was calculated as the difference in the amount of glucose before and after hydrolysis. Glucose was determined via the glucose oxidase method using the glucose autokit (Wako, cat# 439-90901) as per the manufacturer's instructions. Starch was enzymatically digested to glucose by amylase and amyloglucosidase (Megazyme, cat# K-TSTA) by following the product manual and was quantified using the method described above.

**Examination of pollen tube germination and growth**. For the in vitro germination test, pollen grains collected from the dehiscent anthers were placed on germination medium [1.6 mM H$_3$BO$_3$, 1.8 mg Ca(NO$_3$)$_2$, 8.5% sucrose, 0.7% agarose] and incubated at room temperature for 2 h. Then, they were observed under an optical microscope.

For the in vivo germination test, 2–4 h after artificial pollination, rice pistils were subjected to fixation using a fixative (ethanol: acetic acid = 3:1) for 30 min and then mixed with 1 N KOH, and incubated at 55 °C for 30 min. The fixed rice

pistils were subjected to washing steps using distilled water and stained with 0.1% aniline blue in $K_3PO_4$ buffer (pH 8.5), at 25 °C for 2 h. Then, rinsed pistils were visualized under a UV microscope.

**DNA sequence analysis, collection of GA3ox1 sequences, and phylogenetic analysis**. For examination of the *GA3ox1* nucleotide sequence in wild rice, genomic DNA extraction was performed using leaves or samples obtained from the National Institute of Genetics supported by the National Bioresource Project, MEXT, Japan. The DNA fragments of part of the wild rice *GA3ox1* gene were amplified using PCR, using KOD FX Neo (TOYOBO, cat# KFX-201) and the primers listed in Supplementary Table 3. The *GA3ox1* DNA fragments were sequenced using primers considered for PCR.

For tetraploid *Oryza* wild-rice *GA3ox1* orthologs (*O. latifolia*, *O. alta*, *O. grandiglumis*, *O. longiglumis*, and *O. ridleyi*), sequences of two *GA3ox1* orthologs of *O. ridleyi* were obtained; these two orthologs were identical in the region shown in Fig. 3a. For other tetraploid *Oryza* wild rice, the GA3ox1 orthologs could not be sequenced. To ensure that the sequences were orthologous for the *GA3ox1* gene, alignment analysis and phylogenetic analysis were performed with MAFFT version 7.3 using the L-INS-i model[37] program for alignment and phylogenetic tree illustrations were generated with the NJ method by using the GENETYX-Mac (https://www.genetyx.co.jp) software (version 19).

For GA3ox protein sequence alignment and phylogenetic analysis, GA3ox amino acid and nucleotide sequences of various plants listed in Supplementary Table 2 were subjected to a best-BLAST match search in databases and were aligned with MAFFT version 7.3 using the L-INS-i model[37]. For the construction of an amino acid phylogenetic tree, Bayesian estimations of phylogenetic topology were conducted using the ML method by MEGA (ver. 10) with the general time-reversible (+I + G) model, which was selected in ProtTest 3.4.2[38].

**GA binding and yeast two-hybrid assays**. For performing the yeast two-hybrid assay, the Matchmaker Two-Hybrid System (Clontech Laboratories, Inc., cat# 630489) was used. The yeast strain HA109 was used, and the construct was prepared as per methods described previously[1] GID1 in pGBKT7 served as the bait, and SLR1 in pGADT7 was considered as the prey. Plate assays (-His) were performed according to the manufacturer's instructions with modifications to the plates containing $GA_1$, $GA_3$, $GA_4$, and $GA_7$ concentrations ranging between $10^{-10}$ M and $10^{-5}$ M. Control plates contained no GA.

**RNA isolation and real-time RT-PCR analysis**. For RNA isolation from rice and wild rice, the RNeasy plant mini kit (Qiagen, cat# 74904) was used according to the manufacturer's instructions. First-strand cDNA was synthesized from 10 ng or more of total RNA using the Omniscript reverse-transcription kit (Qiagen, cat# 205111). For Real-time RT-PCR analysis, the KOD SYBR qPCR Mix (TOYOBO, cat# QKD-201) and the primers listed in Supplementary Table 3 were used with the CFX96 real-time PCR detection system (Bio-Rad). *Ubiquitin-5* served as the internal control for the normalization of expression levels.

**Expression and purification of OsGA3ox1 and OsGA3ox2**. For the production of the OsGA3ox1 and OsGA3ox2 proteins, OsGA3ox1/pGEX6P1 or OsGA3ox2/pGEX6P1 constructs were overexpressed in *Escherichia coli* Rosetta (DE3) pLysS (Novagen). Cells were grown until the optical density reached 0.8–1.0 $A_{600}$ at 37 °C in Terrific-Broth medium containing 100 μg ml⁻¹ ampicillin, followed by IPTG induction at 18 °C for 16–18 h. The harvested cells were subjected to washing steps using buffer A [10 mM Na-phosphate, 150 mM NaCl, 1 mM DTT (pH 7.5)] containing complete protease inhibitor (Merck) and 1.5 mM phenylmethylsulfonyl fluoride and were then disrupted by sonication. OsGA3ox1 or OsGA3ox2 protein was obtained from the supernatant fraction. The recombinant protein in the supernatant was purified using Glutathione Sepharose 4B resin (Cytiva) equilibrated with buffer A. The column was washed with 30-bed volumes of buffer A and was then eluted with buffer A (plus 20 mM glutathione pH 7.5). To remove the GST-tag, precision protease (Cytiva) was added and incubated overnight at 4 °C. The sample was filtered through a 0.22-μm nylon filter and further purified using a Superdex 200 16/60 column (Cytiva) with buffer A used as a running buffer. The fractions containing the GST and tag-digested OsGA3ox1 or OsGA3ox2 complex were collected and passed through Glutathione Sepharose 4B resin to remove GST. The sample was concentrated and dialyzed with buffer C (10 mM sodium phosphate, pH 7.4) using an Amicon Ultra-4 concentrator unit (10 kDa considered as the molecular weight cut-off) (Millipore). The protein concentrations of OsGA3ox2 were measured by determining UV absorbance at 280 nm. The $A^{0.1\%}$ values at 280 nm were calculated based on the amino acid composition data[39]. The target proteins were detected by comparing with protein standard markers using 12% wt/vol SDS gel, and the optimum conditions for in vitro expression were determined.

**Crystallization**. Crystallization was performed by the sitting drop vapor diffusion method at 20 °C using a mixture containing 1 μl protein and 1 μl mother liquid solution. Droplets containing 20 mg ml⁻¹ OsGA3ox2 dissolved in 5 mM Na-phosphate (pH 7.5), 10 mM 2-oxoglutarate, and 0.5 mM $GA_9$ and mother liquor, were equilibrated against 50 μl of reservoir solution containing 0.15 M ammonium

sulfate, 0.1 M sodium HEPES (pH 7.0), and 20% (w/v) PEG4000. Crystals were flash-frozen in liquid nitrogen with the addition of 20% glycerol in the crystallization mother liquid as a cryoprotectant. X-ray diffraction data were collected using the BL26B1 beamline at SPring-8. All data were processed and scaled using HKL2000[40]. The crystal data are summarized in Supplementary Table 1.

**Structure determination and refinement**. The structure of OsGA3ox2 was determined by performing molecular replacement with Molrep[41,42] using the refined structure and OsGA2ox3 (6KU3) as the search model. Model building and refinement were performed using WinCoot (version 0.8.9)[43], and restrained refinement was performed using REFMAC5[41] from the CCP4 package (version 7) and PHENIX program (version 1.17.1)[44]. The statistically analyzed data after collection and refinement are shown in Supplementary Table 1. Molecular graphics of OsGA3ox2 and predicted models by Alphafold2 were illustrated using PyMOL (http://www.pymol.org).

**GA quantification**. Extraction and semi-purification of GAs were performed as previously described[45,46]. Quantification of GAs was carried out using an ultra-high performance liquid chromatography (UHPLC)-ESI quadrupole-orbitrap mass spectrometer (UHPLC/Q-Exactive™; Thermo Scientific) as described previously[46,47] with an ODS column (AQUITY UPLC HSS T3, 1.8 mm, 2.1 × 100 mm; Waters).

**In situ staining of invertase activity**. Wash buffer [500 μl; 200 mM HEPES-KOH (pH 7.0), 3 mM $MgCl_2$, 1 mM EDTA, 2% glycerin] was added to the anthers, and the anthers were broken in tubes. These were centrifuged at $8000 \times g$ for 5 min at room temperature. Fresh 500 μl wash buffer was added to the precipitate, mixed by vortexing, and centrifuged again at $8000 \times g$ for 5 min at room temperature to remove as much as possible of the material other than pollen. This was repeated once more to remove the soluble sugars. These were resuspended in wash buffer and dispensed in 20 μl portions, and Incubation buffer [20 U ml⁻¹ glucose oxidase, 0.014% phenazine methosulphate, 0.024% nitroblue tetrazolium salt, 0.5% Sucrose], adjusted to each pH using 100 mM citric acid and 200 mM $K_2HPO_4$, was added. Sucrose was not added to the Incubation buffer in the control experiment. The supernatants were removed by centrifugation at $8000 \times g$ for 5 min at room temperature after shaking for 16 h at 26 °C in the dark. Then, 200 μl 70% ethanol was added, and the ethanol was removed by centrifugation at $8000 \times g$ for 5 min at room temperature. The pollen grains were suspended in 20 μl of sterile water and observed and photographed under an optical microscope.

**DAPI and FDA staining**. For DAPI staining, anthers were washed with 80% ethanol after fixation, placed on a glass slide, and 5 μl 2 μg ml⁻¹ DAPI staining solution (2 μl of 1 mg ml⁻¹ DAPI scale-up to 1 ml in PBS) was added. The anthers were crushed in DAPI solution. Pollen was placed on a glass slide and covered with a cover glass for observation.

For FDA staining, a 10 μl drop of 0.5 μg ml⁻¹ FDA solution (0.5 mg ml⁻¹ FDA in DMSO diluted 1000-fold with PBS solution) was placed on a glass slide, and two anthers were crushed in the solution to release the pollen. As a negative control, PBS solution containing pollen was heated in a block incubator at 100 °C for 5 min, and then 0.5 μg ml⁻¹ FDA solution was added. After covering with a cover glass, the cells were incubated at room temperature for 5 min and observed with a ZEISS LSM 700 confocal microscope.

**Anther fraction isolation and invertase enzyme assays**. For anther fraction, randomly collected rice anthers (each sample contained approximately 250 pollen grains) were sonicated in 5 mM sodium phosphate buffer (pH 7.0), centrifuged at $15,000 \times g$ for 30 min at 4 °C to separate them into soluble invertase (supernatant) (A fraction in Fig. 2i and j) and wall-bound invertase (precipitate). Furthermore, 1.3 M NaCl was added to the precipitated fraction, incubated overnight, separated into the supernatant, and precipitated by centrifugation (D and E fractions in Fig. 2i, j). Each sample, including the one incubated overnight in 5 mM phosphate buffer as a control (B and C fractions in Fig. 2i, j), was used for subsequent enzyme test.

For invertase enzyme assays, invertase activity was calculated according to a previously described protocol[48,49], with some modifications. For each sample, 50 μL of solution were diluted in 0.2 M sodium acetate buffer (pH 5.5 for wall-bound invertase assays) to a final volume of 600 μL. The enzymatic reaction was triggered by adding 800 μL of 0.225 M sucrose. An aliquot of 700 μL ($t_{1 h}$ sample) was transferred to a new tube, whereas the remaining volume ($t_{0 h}$ sample) was mixed with 200 μL Somogyi's solution (Sigma, cat# 28-4730) and heated for 15 min at 100 °C. The $t_{1 h}$ sample was incubated at 30 °C for 1 h, and the reaction was stopped as described before. Samples were then placed on ice for 5 min, and 200 μL Nelson's solution (Sigma, cat# 21-0925) was added and left for 5 min. After 20 min, the absorbance at 660 nm was measured. The invertase activity was calculated as the difference in reduced sugar levels between $t_{1 h}$ and $t_{0 h}$ and was represented as μg of glucose formed in 1 ml of extract per hour and mg of total protein.

For western blotting, randomly collected rice anthers (each sample contained approximate three thousand pollen grains) were used. After adding 2× sample buffer to the protein fractions, the samples were boiled for 5 min. Protein samples

were separated by 6% SDS-PAGE and transferred to a PVDF membrane using the Trans-Blot Turbo transfer system (Bio-Rad Laboratories, cat #1704156). To detect OsCIN3 protein, blots were blocked with 5% skim milk in TBST buffer [0.1% Tween 20 in 2 mM Tris-HCl, pH 7.6, 13.7 mM NaCl] for 1 h, followed by incubation with rabbit-grown anti-OsCIN3 antiserum (1:5,000 dilution, Cosmo Bio) at 4 °C overnight. The blots were washed three times with TBST buffer for 15 min each. The membrane was incubated with goat anti-rabbit IgG horseradish peroxidase-conjugated secondary antibody (1:10,000 dilution) for 45 min, and then washed following the same procedure above. All reactions, except the primary antibody reaction, were performed at room temperature.

**Statistics and reproducibility**. Results are presented as the mean ± SD. Statistical tests were selected on a case-by-case basis, depending on the nature of the data collected. More information on the selection of tests can be found in the figure legends. A two-tailed probability of less than 0.05 was considered statistically significant.

**Reporting summary**. Further information on research design is available in the Nature Research Reporting Summary linked to this article.

## Data availability
The coordinates were deposited into the Protein Data Bank with accession number 7EKD. All source data in this study are included in Supplementary Data 1. The results of the statistical analyses performed in this study are included in Supplementary Data 2. The sequencing data are included in Supplementary Data 3. The uncropped images of the western blot and CBB-stained gel in Fig. 2 are included in the Supplementary Information file as Supplementary Figure 13.

## Code availability
The code used in this work is included as Supplementary Code 1 and is also available on the Zenodo (https://doi.org/10.5281/zenodo.5748638)[50].

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

## Acknowledgements

This work was supported by Grants-in-Aid for Scientific Research on Innovative Areas [grants number, 16H06464 (to M.U.-T.) and 16H06468 (to M.U.-T. and M.M.)], a Grant-in-Aid for Scientific Research (B) [Grant number, 16H04907, and 20H03272 (to M.U.-T.)], and "Graduate Program of Transformative Chem-Bio Research" in Nagoya University, supported by MEXT (WISE Program) (to K.K.). We thank M. Endo (National Agriculture and Food Research Organization (NARO), Japan) for providing the CRISPR–Cas9 vectors. The wild rice accessions used in this study were distributed by the National Institute of Genetics supported by the National Bioresource Project (NBRP), AMED, Japan.

## Author contributions

S.T. and M.U.-T. conceived and designed the project; K.K. and T.K. performed construct design, ensured provision of proper care to the transgenic plant in the greenhouse, and analyzed phenotypes. T.K. and S.T. performed purification, crystallization, and structure determinations. M.Morii performed the evaluation of starch and sugar content and conducted the Y2H assay. A.S. provided assistance with the protein purification. H.Y. provided assistance with enzyme assays and ensured the provision of proper care to the transgenic plants. I.A. provided assistance with RNA extraction, expression analyses, and transformation. H.M. provided assistance with the Y2H assay. Y.Toda provided assistance with the quantification of I2-KI staining results. M.K., Y.Takebayashi, and H.S. quantified hormone levels. H.F. and K.N. provided assistance with the growth of wild rice. B.M. helped to solve and refine the structure. T.A. provided assistance with the evolutionary analyses. H.K and M. Matsuoka provided assistance with and advised on scientific writing. K.K., M.Morii, S.T., and M.U.-T. wrote the manuscript.

## Competing interests

The authors declare no competing interests.
