## [Peer Review File · Communications Biology]

Reviewers' comments:

Reviewer #1 (Remarks to the Author):

The manuscript by Kawai et al reports an association between the expression and the activity of specific alleles of GA 3 oxidase 1 that emerged at one point in the evolution of Poaceae. It is known that 3ox1 can synthesize both GA1 and GA7, and they find that rice OsGA3ox1 is particularly good at making GA7, compared with GA3ox2 (which mostly makes GA4). Interestingly, they also find that this particular OsGA3ox1 is specifically expressed in anthers, and the Osga3ox1 mutant has defects in pollen development. Then they examine the presence of the specific allelic change (Y to F) that allows OsGA3ox1 to produce so much GA7, and they see that only a recent clade of *Oryza* species have the same allele, and the gene also shows a tendency to be preferentially expressed in anthers in those species. Based on the association between high GA7 production, and expression restricted to anthers, they propose that the two characteristics are the consequence of GA7 being an active GA that cannot be inactivated by C2-oxidation: accumulation of this active GA in vegetative tissues would have a deleterious effect in growth, so it can exist only thanks to being expressed only in anthers. The idea is stimulating, and the authors provide several circumstantial observations and correlations pointing in that direction. However, in my opinion, there are two weak points at the moment:

1- This cannot be considered as an "evolutionary arms race".

2- The idea that a GA7-producing GA3ox1 enzyme is necessary in anthers and deleterious in vegetative tissues is not yet supported by strong evidence.

1. As the authors indicate, "evolutionary arms race" is applied to species that fight one another for their survival. Genes are involved in this process, of course, but opposing interests must exist between the activity of those genes, and this is not the case in the process studied here. GA synthesis does not require to diminish GA inactivation or vice-versa. What the authors describe is a case of co-adaptation, by which the appearance of a mutation is compensated by a second one in the same (or a different) gene. It is not clear how things happened in evolution, but a likely event (following the authors' idea) is that 3ox1 became anther specific, and this allowed the appearance of a Y-to-F mutation that allowed increased production of GA7. Or perhaps it was the other way around. In any case, there are multiple examples in the literature of this sort of compensation.

2. The authors do not demonstrate the premise that higher GA7 production in shoots would cause fitness defects -as the main cause for restricting the expression of that new GA3ox1 to anthers. What they show is correlations between the expression pattern and the GA7/GA4 ratios in several Poaceae.

a) Why is *O. granulata* the only species with an "F"-GA3ox1 enzyme tested? For the correlations to be stronger, it would have been desirable to test *O. meyeriana*, *O. longiligumis* and/or *O. ridleyi*, for instance

b) The complementation with the GA3ox1 locus of "F" and "Y" species is a nice experiment, but it involves transcriptional effects and changes in activity at the same time. And only the anther development phenotype is shown -not the phenotype of the rest of the plant, and its fitness.

c) A more direct test would involve complementing the Osga3ox1 mutant with the OsGA3ox1 mutated (F back to Y), to see if GA7 is indeed "necessary" for anther development (my guess is that GA4 will do a very good job. This result would imply that there is no "need" to produce GA7 in anthers; it may be by chance, or it may be because its production should not occur in other parts of the plants)

d) Finally, what should be shown is the effect of expressing the "F"-OsGA3ox1 in the domain of expression of OsGA3ox2 and check fitness (if the authors are right, there should be a penalty).

Finally, there are other minor issues that could be solved in the text:

- Line 28 "we report that evolutionary forces between phytohormone synthesis and inactivating enzymes had been occurred in wild rice". The English language is not optimal (actually it should be edited throughout the text) and the idea is not correct: "evolutionary forces between synthesis and inactivation" is not what is happening here.

- Line 31 "GA is transcriptionally and post-transcriptionally regulated". Perhaps "GA homeostasis is... regulated"

- Line 40 "when the AA species emerged". It is not obvious, for a wide readership, what "AA" species are. This should be explained in a different way here in the abstract.

- Line 52: "The GID1 receptor and GA synthetic enzymes were developed simultaneously in fern". This is quite wrong from an evolutionary perspective. The GA/GID1 perception system developed in the ancestor of vascular plants; ferns are only one extant lineage within tracheophytes, and they did not "convert" later to gymnosperms and angiosperms. Rather, all those lineages are sisters

- Line 54: "evolved in gymnosperms" (same wrong concept as above)

- I would reconsider reducing the description of the Osga3ox1 mutant phenotypic defects, the connection with invertase genes, etc. It is not completely central to the story, and it is not evident that starch defects are the main cause of male sterility.

- Line 163: "Together, we concluded that a high abundance of non-13 hydroxyl bioactive GAs, especially GA7 produced by OsGA3ox1 in pollen, played an important role...". Why "especially GA7"? As discussed in point 2, it is not demonstrated that this is done through GA7. It could be GA4 AND GA7. The likely higher GID1-binding activity of GA7 may make it quantitatively more important despite the lower level compared with GA4, but this is only hypothetical.

- Lines 248-258 (Discussion) - again, I would suggest rewriting it to remove the wrong evolutionary descriptions ("was established in fern") and the arms-race idea.

Reviewer #2 (Remarks to the Author):

Kawai et al. characterize a rice (*Oryza sativa*) isoform of the gibberellin (GA) biosynthesis enzyme GA3 oxidase (GA3ox1) that is specifically expressed in anthers. After discovering that rice anthers contain a relatively high amount of the compound GA7, a GA that is resistant to catabolization by GA degrading enzymes, they use a rice mutant and enzyme activity measurements to show that GA3ox1 is responsible for the synthesis of GA7 in anthers. They also determine the crystal structure of the enzyme and identify that a Tyr-to-Phe substitution that occurred during evolution of the genus *Oryza* is responsible for the distinct characteristics of this enzyme. They show that GA3ox1 is required for starch accumulation, pollen viability and fertilization, and they postulate that the ability to synthesize GA7 conferred an advantage for rice reproduction and dissemination. The work describes a molecular event that helped to shape the developmental characteristics of cultivated rice during evolution.

Specific comments:

-Abstract, line 40: Pollen tube germination was not measured (only elongation).

-Line 81: The experiment in yeast does not show the "abundance" of GA7. It shows the "capacity" or "efficiency" of GA7 to promote the formation of the GID-DELLA complex.

-Lines 148-165: The results shown in Ext. Data Fig. 7 and in Fig. 2 do not show that OsCIN3/OsCIN4 is responsible for the decrease in invertase activity observed in *ga3ox1* mutant pollen. In fact, no difference in transcript levels for this enzyme were observed in the mutant relative to the wild-type. Other invertases are expressed in pollen and it is possible that these are involved. The sentence in line 165 stating that GA3ox1 acts "via upregulation of OsCIN3/OsINV4" should be removed or supported with further experimental evidence.

-Fig. 3c: How do the authors explain that *L. perrieri* has a similar GA7/GA4 ratio as species with Phe-GA3ox1?

-Fig. 1a: Clarify the meaning of the bars with different colours (wt vs mutant).

-Fig. 2: Check for misspellings in the graphs. "steined" "sucurose", for example

-In experiments with transgenic lines, please indicate how many independent lines were analyzed for each construct. Was the expression of transgenes analyzed in the different lines? Expression levels may depend on the insertion site and may vary in different lines with the same construct.

-In the same way, clarify if the phenotypic analysis of the CRISPR ga3ox1 mutant was done in one or more independent lines. Using more than one line is recommended to rule out that the effects are due to modifications in other regions of the genome.

-Ext Data Fig. 1a: Please, indicate the synthesis of GA3 and GA7 also in the scheme

-Careful language editing is required: line 29 "had been occurred"; line 36 "is likely account"; line 204 "pettern"; line 299 "this experiments"; lines 336, 342 "straining" and other parts of the manuscript.

We thank the two reviewers very much for their detailed comments and suggestions on our manuscript. Below are detailed point-by-point responses to the comments and issues pointed out. We hope that the responses below address all the comments and suggestions.

We marked the parts of the manuscript that we have revised with a yellow marker.

RESPONSE TO REVIEWER #1:

The manuscript by Kawai et al reports an association between the expression and the activity of specific alleles of GA 3 oxidase 1 that emerged at one point in the evolution of Poaceae. It is known that 3ox1 can synthesize both GA1 and GA7, and they find that rice OsGA3ox1 is particularly good at making GA7, compared with GA3ox2 (which mostly makes GA4).

Interestingly, they also find that this particular OsGA3ox1 is specifically expressed in anthers, and the Osga3ox1 mutant has defects in pollen development. Then they examine the presence of the specific allelic change (Y to F) that allows OsGA3ox1 to produce so much GA7, and they see that only a recent clade of Oryza species have the same allele, and the gene also shows a tendency to be preferentially expressed in anthers in those species.

Based on the association between high GA7 production, and expression restricted to anthers, they propose that the two characteristics are the consequence of GA7 being an active GA that cannot be inactivated by C2-oxidation: accumulation of this active GA in vegetative tissues would have a deleterious effect in growth, so it can exist only thanks to being expressed only in anthers. The idea is stimulating, and the authors provide several circumstantial observations and correlations pointing in that direction. However, in my opinion, there are two weak points at the moment:

1- This cannot be considered as an “evolutionary arms race”.

2- The idea that a GA7-producing GA3ox1 enzyme is necessary in anthers and deleterious in vegetative tissues is not yet supported by strong evidence.

Thanks to Reviewer #1 for deeply understanding our manuscript and for the helpful comments. We agree with Reviewer #1's comments about the two weak points and appreciate pointing them out. Regarding weak point 1, we revised the introduction and conclusion and corrected the misconceptions. Regarding weak point 2, we conducted additional experiments to support our conclusions.

Q1:

As the authors indicate, “evolutionary arms race” is applied to species that fight one another for

their survival. Genes are involved in this process, of course, but opposing interests must exist between the activity of those genes, and this is not the case in the process studied here. GA synthesis does not require to diminish GA inactivation or vice-versa.

What the authors describe is a case of co-adaptation, by which the appearance of a mutation is compensated by a second one in the same (or a different) gene. It is not clear how things happened in evolution, but a likely event (following the authors' idea) is that 3ox1 became anther specific, and this allowed the appearance of a Y-to-F mutation that allowed increased production of GA7. Or perhaps it was the other way around. In any case, there are multiple examples in the literature of this sort of compensation.

R1:

We thank Reviewer #1 for the comments. According to the suggestion, we have removed the statements referring to an evolutionary arms race in the Introduction and the Results and Discussion sections. In the first version of the manuscript, we meant to write “evolutionary arms race” in reference to the evolution of gibberellin synthase and metabolizing enzymes in land plants, but we have not been able to provide such evidence directly in this study in our first manuscript. We regret this oversight.

In addition, in the Introduction section, we have added sentences that provide readers with the research background of this paper, such as the regulation of GA synthase and metabolizing enzymes in plants and the function of GAs in flowers, and we have highlighted that too much GA has a negative effect on shoot development. By citing the example of “Bakanae disease,” a fungal disease of rice characterized by the synthesis of large amounts of GA₃, we described how excessive amounts of bioactive GAs affect shoot elongation (Tudzynski and Höflter, 1998; Gupta et al., 2015). Similar to GA₇, GA₃ is a bioactive form of GA that is not inactivated by the rice GA inactivating enzyme GA2oxs. The statements about the Bakanae fungus will make the reader more aware of the importance of regulating endogenous GA levels in shoots.

In the Results and Discussion section, we removed the statements on arms race and described how *Oryza* plants have evolved the expression and the enzymatic activity of *GA3ox1*. We explained that bioactive GAs produced by *GA3ox1* function in anthers and play an important role in reproduction, contributing to improved fertility of rice.

To reflect the revisions made in the manuscript, we have revised the title of the manuscript as follows:

Evolutionary alterations in gene expression and enzymatic activities of gibberellin 3-oxidase 1 in *Oryza*

Q2:

The authors do not demonstrate the premise that higher GA7 production in shoots would cause fitness defects -as the main cause for restricting the expression of that new GA3ox1 to anthers. What they show is correlations between the expression pattern and the GA7/GA4 ratios in several Poaceae.

R2

We hope that we have addressed Reviewer #1's question by answering questions a–d below.

Q2a:

*Why is *O. granulata* the only species with an “F”-GA3ox1 enzyme tested? For the correlations to be stronger, it would have been desirable to test *O. meyeriana*, *O. longigiumis* and/or *O. ridleyi*, for instance*

R2a:

It seems that the “F”-GA3ox1 enzyme pointed out by Reviewer #1 is the “Y”-GA3ox1 enzyme. As Reviewer #1 suggested, we performed additional measurements of enzymatic activities of GA3ox1 from other GG genome plants of two accessions with Y-type GA3ox1 in their genome, *O. meyeriana* (Accession No. W1360 and W2068), as shown below. They also showed that GA₇ synthesis activity is lower than in the F-type ones (**a** in the Figure below). Gene expression was shown in organs other than anthers (stems and whole spikelets) (**b**).

Enzyme activity and gene expression of *GA3ox1* in *O. meyeriana*

However, we could not measure the GA₇ productivity using GA_{3ox1} enzymes of *O. longigiumis* or *O. ridleyi* because of the following reasons:

1. Since The HHJJ genome plants, *O. longigiumis* and *O. ridleyi* are tetraploids, two orthologs of *GA3ox1* may exist in the genome with similar sequences, making it difficult to isolate the full length CDS.
2. The genomic information of the HHJJ genomic plants has not been elucidated, and the exact gene sequence of *GA3ox1* in them is unknown. This might be partly due to the fact that, with regard to the HH and JJ genomes, which are necessary to elucidate the genomic information of HHJJ genome plants, diploid plants with those genomes are already extinct (Ge et al., 1999).

Q2b:

The complementation with the *GA3ox1* locus of “F” and “Y” species is a nice experiment, but it involves transcriptional effects and changes in activity at the same time. And only the anther development phenotype is shown -not the phenotype of the rest of the plant, and its fitness.

R2b:

We have analyzed the *in vitro* and *in vivo* enzymatic activity of GA_{3ox1} in wild rice (Figure 3c and

4h). Gene expression was also analyzed using wild rice (Figure 4a–g). We designed this complementation experiment to compare the effects of both enzyme activity and expression simultaneously on the same background (*osga3ox1* mutant). Therefore, in this experiment, to observe the effect of *GA3ox1* in planta, we had to introduce the region of *GA3ox1*, including the promoter, into the plant. We have added a description of the purpose of this experiment in the revised manuscript (line 325-326).

We analyzed their gene expression, analyzed the phenotype of plant height, quantified the amount of endogenous GA in the anthers, and added the results in the revised manuscript. These results and Figures 5d and e show that in the same *osga3ox1* background, plants transformed with wild rice *GA3ox1*, which has a higher capacity for GA₇ synthesis, showed a greater recovery in reproductive capacity. Conversely, none of the transformants had any effect on vegetative growth. We added these data as Supplementary Figure 12d–g in the revised version of the manuscript.

Q2c:

*A more direct test would involve complementing the *Osga3ox1* mutant with the *OsGA3ox1* mutated (F back to Y), to see if GA₇ is indeed “necessary” for anther development (my guess is that GA₄ will do a very good job. This result would imply that there is no “need” to produce GA₇ in anthers; it may be by chance, or it may be because its production should not occur in other parts of the plants)*

R2c:

Instead of conducting the experiment to complement the *osga3ox1* mutant with the *OsGA3ox1* mutated (F back to Y) and to add additional support to the conclusion that GA₇ is required for anther development, we measured the amount of endogenous GAs in anthers of plants used in the complementation experiment (Supplementary Figure 12g in the revised manuscript). We believe that GA quantification, enzyme activity of wild rice *GA3ox1* (Figure 3c and 4h), and phenotypic recovery of *osga3ox1* mutant by GA treatment (Figure 2c) are sufficient to demonstrate that anthers require large amounts of GA₇ in addition to GA₄. Considering the fact that AA genome plants and transformants induced *GA3ox1* of AA plants show a higher KI staining and reproductive success rate than others (Figure 5d and e, Supplementary Figure 12b and c in the revised manuscript), it might be necessary for the anthers to synthesize large amounts of GA₇.

Q2d:

*Finally, what should be shown is the effect of expressing the “F”-*OsGA3ox1* in the domain of expression of *OsGA3ox2* and check fitness (if the authors are right, there should be a penalty).*

R2d:

Thank you for suggesting this experiment. Since the *OsGA3ox2* promoter is subject to negative feedback regulation by the amount of endogenous GAs (Itoh, H. et al., 2001), we instead showed the fitness of the bioactive GAs synthesized by *GA3ox1* in the shoot by direct GA treatment in the shoot of the plants. As shown in Supplementary Figure 1c and d in the revised version of the manuscript, GA, especially GA₃ and GA₇, which do not undergo inactivation, induced more pronounced rice shoot growth. In addition, the *slr1* mutant, which is under a continuous GA response, began to wither 13 days after germination. Besides, in the Introduction section, we cited papers on “Bakanae diseases” (Gupta et al., 2015) that infect rice and produce excessive GA and other papers that show the importance of regulating endogenous GA production for plants to make it easier for readers to understand.

Supplementary Figure 1c and d

Finally, there are other minor issues that could be solved in the text:

Q3:

- Line 28 “we report that evolutionary forces between phytohormone synthesis and inactivating enzymes had been occurred in wild rice”. The English language is not optimal (actually it should be edited throughout the text) and the idea is not correct: “evolutionary forces between synthesis and inactivation” is not what is happening here.

R3:

According to the comment, we have removed the statements referring to an evolutionary arms race in the manuscript. Instead, we added the sentence as described in **R1** above. In addition, after careful checking, we used an English review service to correct language-related mistakes.

Q4:

- Line 31 *“GA is transcriptionally and post-transcriptionally regulated”*. Perhaps *“GA homeostasis is... regulated”*

R4:

Thank you for pointing this out. We rewrote the Abstract.

Q5:

- Line 40 *“when the AA species emerged”*. It is not obvious, for a wide readership, what *“AA”* species are. This should be explained in a different way here in the abstract.

R5:

Thank you for the suggestion. We rewrote the Abstract.

Q6:

- Line 52: *“The GID1 receptor and GA synthetic enzymes were developed simultaneously in fern”*. This is quite wrong from an evolutionary perspective. The GA/GID1 perception system developed in the ancestor of vascular plants; ferns are only one extant lineage within tracheophytes, and they did not *“convert”* later to gymnosperms and angiosperms. Rather, all those lineages are sisters

R6:

Thank you for pointing this out. We rewrote the manuscript and deleted these parts in the revised version.

Q7:

- Line 54: *“evolved in gymnosperms”* (same wrong concept as above)

R7:

Thank you for pointing this out. We rewrote the text.

Q8:

- I would reconsider reducing the description of the *Osga3ox1* mutant phenotypic defects, the connection with invertase genes, etc. It is not completely central to the story, and it is not evident

that starch defects are the main cause of male sterility.

R8:

Thank you for the suggestion. As pointed out by the Editor and Reviewer #2, we have added western blot data for invertase (Figure 2k in the revised version of the manuscript). This result complements the relationship between GAs and invertase. We also believe that these results and explanations are necessary to improve our manuscript.

Western blot analysis of the anther extract fractions using an OsCIN3-specific antibody

Q9:

- Line 163: “Together, we concluded that a high abundance of non-13 hydroxyl bioactive GAs, especially GA₇ produced by *OsGA3ox1* in pollen, played an important role....”. Why “especially GA₇”? As discussed in point 2, it is not demonstrated that this is done through GA₇. It could be GA₄ AND GA₇. The likely higher GID1-binding activity of GA₇ may make it quantitatively more important despite the lower level compared with GA₄, but this is only hypothetical.

R9:

Thank you for pointing this out. We changed the sentence “non-13 hydroxyl bioactive GAs, especially GA₇ produced by *OsGA3ox1*” to “GA₄ and GA₇ produced by *OsGA3ox1*” (line 230). GA₇ has higher GID1-binding activity (Supplementary Figure 1f) and is not inactivated by the inactivating enzyme GA2oxs. We believe that *OsGA3ox1* synthesizes large amounts of GA₇ as well as GA₄ in the anther and that these large amounts of GAs are necessary for the activity of invertases. GA₄ also works, but it might be necessary for rice to synthesize GA₇ as well to increase the total endogenous amount of bioactive GAs. This can be supported by the correlation between endogenous GA₇ and starch content shown in Figure 5c.

Q10:

- Lines 248-258 (Discussion) - again, I would suggest rewriting it to remove the wrong evolutionary

descriptions (“was established in fern”) and the arms-race idea.

R10:

Thank you for pointing this out. We rewrote the text.

References

Tudzynski, B. & Höltner, K. Gibberellin biosynthetic pathway in *Gibberella fujikuroi*: evidence for a gene cluster. *Fungal Genet. Biol.* **25**, 157-170 (1998).

Gupta A, Solanki I, Bashyal B, Singh Y, Srivastava K. Bakanae of rice- an emerging disease in Asia. *JAPS, J. Animal and Plant Sci.* **25**, 1499-1514 (2015).

Ge, S., Sang, T., Lu, B. R. & Hong, D. Y. Phylogeny of rice genomes with emphasis on origins of allotetraploid species. *Proc. Natl Acad. Sci. USA* **96**, 14400–14405 (1999).

Itoh, H. et al. Cloning and functional analysis of two gibberellin 3 beta -hydroxylase genes that are differently expressed during the growth of rice. *Proc. Natl Acad. Sci. U. S. A.* **98**, 8909–8914 (2001).

RESPONSE TO REVIEWER #2:

Kawai et al. characterize a rice (Oryza sativa) isoform of the gibberellin (GA) biosynthesis enzyme GA3 oxidase (GA3ox1) that is specifically expressed in anthers. After discovering that rice anthers contain a relatively high amount of the compound GA7, a GA that is resistant to catabolization by GA degrading enzymes, they use a rice mutant and enzyme activity measurements to show that GA3ox1 is responsible for the synthesis of GA7 in anthers.

They also determine the crystal structure of the enzyme and identify that a Tyr-to-Phe substitution that occurred during evolution of the genus Oryza is responsible for the distinct characteristics of this enzyme. They show that GA3ox1 is required for starch accumulation, pollen viability and fertilization, and they postulate that the ability to synthesize GA7 conferred an advantage for rice reproduction and dissemination. The work describes a molecular event that helped to shape the developmental characteristics of cultivated rice during evolution.

Thanks to Reviewer #2 for deeply understanding our manuscript and helpful comments. We have responded to your comments point by point below. Additional experiments were conducted on the relationship between invertase and gibberellin and the cause of the high GA₇ synthesis ratio of *L.*

perrieri GA3ox1, which was not shown in the original manuscript.

Specific comments:

Q1:

-Abstract, line 40: *Pollen tube germination was not measured (only elongation).*

R1:

We performed an experiment with pollen germination and added it in Figure 2e and Supplementary Figure 7h in the revised manuscript.

***In vitro* pollen germination test (Figure 2e and Supplementary Figure 7h)**

Q2:

-Line 81: *The experiment in yeast does not show the “abundance” of GA7. It shows the “capacity” or “efficiency” of GA7 to promote the formation of the GID-DELLA complex.*

R2:

Thank you for pointing this out. We changed the word “abundance” to “efficiency” in line 130 in the revised manuscript.

Q3:

-Lines 148-165: *The results shown in Ext. Data Fig. 7 and in Fig. 2 do not show that*

OsCIN3/OsCIN4 is responsible for the decrease in invertase activity observed in *ga3ox1* mutant pollen. In fact, no difference in transcript levels for this enzyme were observed in the mutant relative to the wild-type. Other invertases are expressed in pollen and it is possible that these are involved. The sentence in line 165 stating that *GA3ox1* acts “via upregulation of *OsCIN3/OsINV4*” should be removed or supported with further experimental evidence.

R3:

Thank you for pointing this out. We added the western blot analysis in Figure 2k in the revised manuscript using an *OsCIN3*-specific antibody (the figure below) on the anther protein fractions containing about 250 pollen grains shown in Figure 2i and j. The data show that the amount of *OsCIN3* protein in anther is similar in total extracts of wild-type and *osga3ox1* mutant. But in cell wall fractions (fraction C and D in Figure 2k in the revised version), *OsCIN3* protein level in *osga3ox1* was lower than that in wild-type; therefore, it indicates that *OsGA3ox1* may be involved in the precise retainment of *OsCIN3* protein to the cell wall. In the revised manuscript, we have added statements about this experiment (line 220-232).

In accordance with Communications biology guidelines, *OsCIN3/OsINV4* has been rewritten as *OsCIN3*.

Epitope design of *OsCIN3*-specific antibody

Western blot analysis of the anther extract fractions using an OsCIN3-specific antibody (Figure 2k)

Q4:

-Fig. 3c: How do the authors explain that *L. perrieri* has a similar GA₇/GA₄ ratio as species with *Phe-GA3ox1*?

R4:

To make this point clearer, we used the AlphaFold2, a deep learning-based program to predict GA3ox1 structures from wild rice and *L. perrieri* and compared them to elucidate why *L. perrieri* GA3ox1 has a similar GA₇/GA₄ ratio to F-type. GA3ox1 from *L. perrieri* has the same residues in the substrate and co-substrate interaction site with Y-type ones; however, these predictions clearly showed that 2OG-interacting Arg315 in *L. perrieri* GA3ox1 is at a different angle from others. Therefore, this structure might weaken the interaction with 2OG like F-type GA3ox1. However, further experimental data are needed to confirm this. These modeling data were added in Supplementary Fig 10 in the revised manuscript.

a

	GA ₃ interaction site						2OG interaction site			Fe (II) interaction site			
	Tyr ⁴²	Ser ²⁵³	Gly ²⁵⁴	Phe ²⁵⁶	Thr ²⁵⁷	Leu ³²⁵	Arg ³²⁰	Phe ²³⁵	Arg ¹⁸	Ser ³²⁰	His ²⁵⁰	Asp ²⁵²	His ³⁰⁸
Oryza sativa _GA3ox1	Y	S	G	F	T	L	R	F	R	S	H	D	H
Oryza nivara _GA3ox1	Y	S	G	F	T	L	R	F	R	S	H	D	H
Oryza rufipogon _GA3ox1	Y	S	G	F	T	L	R	F	R	S	H	D	H
Oryza barthii _GA3ox1	Y	S	G	F	T	L	R	F	R	S	H	D	H
Oryza meridionalis _GA3ox1	Y	S	G	F	T	L	R	F	R	S	H	D	H
Oryza glumipatula _GA3ox1	Y	S	G	F	T	L	R	F	R	S	H	D	H
Oryza punctata _GA3ox1	Y	S	G	F	T	L	R	F	R	S	H	D	H
Oryza australiensis _GA3ox1	Y	S	G	F	T	L	R	F	R	S	H	D	H
Oryza brachyantha _GA3ox1	Y	S	G	F	T	L	R	F	R	S	H	D	H
Oryza granulata _GA3ox1	Y	S	G	F	T	L	R	F	R	S	H	D	H
Oryza sativa _GA3ox2	Y	S	G	F	T	L	R	Y	R	S	H	D	H
Oryza nivara _GA3ox2	Y	S	G	F	T	L	R	Y	R	S	H	D	H
Oryza rufipogon _GA3ox2	Y	S	G	F	T	L	R	Y	R	S	H	D	H
Oryza meridionalis _GA3ox2	Y	S	G	F	T	L	R	Y	R	S	H	D	H
Oryza glumipatula _GA3ox2	Y	S	G	F	T	L	R	Y	R	S	H	D	H
Oryza punctata _GA3ox2	Y	S	G	F	T	L	R	Y	R	S	H	D	H
Oryza brachyantha _GA3ox2	Y	S	G	F	T	L	R	Y	R	S	H	D	H
Oryza granulata _GA3ox2	Y	S	G	F	T	L	R	Y	R	S	H	D	H
Leersia perrieri _GA3ox1	Y	S	G	F	T	L	R	Y	R	S	H	D	H
Leersia perrieri _GA3ox2	Y	S	G	F	T	L	R	Y	R	S	H	D	H
Hordeum vulgare _GA3ox1	Y	S	G	F	T	L	R	Y	R	S	H	D	H
Hordeum vulgare _GA3ox2	Y	S	G	F	T	L	R	Y	R	S	H	D	H
Triticum aestivum _GA3ox-2A	Y	S	G	F	T	L	R	Y	R	S	H	D	H
Triticum aestivum _GA3ox-2B	Y	S	G	F	T	L	R	Y	R	S	H	D	H
Triticum aestivum _GA3ox-2D	Y	S	G	F	T	L	R	Y	R	S	H	D	H
Triticum aestivum _GA3ox-3A	Y	S	G	F	T	L	R	Y	R	S	H	D	H
Triticum aestivum _GA3ox-3B	Y	S	G	F	T	L	R	Y	R	S	H	D	H
Triticum aestivum _GA3ox-3D	Y	S	G	F	T	L	R	Y	R	S	H	D	H
Brachypodium distachyon _GA3ox2a	Y	S	G	F	T	L	R	Y	R	S	H	D	H
Brachypodium distachyon _GA3ox2b	Y	S	G	F	T	L	R	Y	R	S	H	D	H
Oropetium thomaeum _Oropetium_20150105_07284A	Y	S	G	F	T	L	R	Y	R	S	H	D	H
Sorghum bicolor _GA3ox1	L	S	G	I	T	L	K	Y	R	S	H	D	H
Sorghum bicolor _GA3ox2	L	S	G	F	T	L	R	Y	R	S	H	D	H
Zea mays _GA3ox1	L	S	G	I	T	L	K	Y	R	S	H	D	H
Zea mays _GA3ox2	L	S	G	F	T	L	R	Y	R	S	H	D	H
Setaria italica _Si025127m.g	-	S	G	I	T	L	R	Y	R	S	H	D	H
Setaria italica _Si001832m.g	-	S	G	F	T	L	R	Y	R	S	H	D	H
Panicum virgatum _Pavir.J24392.1	-	S	G	F	T	L	R	Y	R	S	H	D	H
Panicum virgatum _Pavir.Eb00358.1	-	S	G	F	T	L	R	Y	R	S	H	D	H
Panicum virgatum _Pavir.J17485.1	-	S	G	F	T	L	R	Y	R	S	H	D	H
Oryza sativa _GA20ox2	A	P	T	L	G	L	T	Y	R	S	H	D	H

Substrate interaction site of *Oryza* GA3ox1 (Supplementary Figure 10).

Q5:

-Fig. 1a: Clarify the meaning of the bars with different colours (wt vs mutant).

R5:

We clarified the meaning of the bars with different colors.

Q6:

-Fig. 2: Check for misspellings in the graphs. "steined" "sucurose", for example

R6:

Thank you for your comments. We carefully checked and corrected the misspellings.

Q7:

-In experiments with transgenic lines, please indicate how many independent lines were analyzed for each construct. Was the expression of transgenes analyzed in the different lines? Expression levels may depend on the insertion site and may vary in different lines with the same construct.

R7:

We used two independent lines each for our experiments. As Reviewer #2's suggested we have added qRT-PCR data in Supplementary Figure 12d.

Supplementary Figure 12d

Q8:

-In the same way, clarify if the phenotypic analysis of the CRISPR *ga3ox1* mutant was done in one or more independent lines. Using more than one line is recommended to rule out that the effects are due to modifications in other regions of the genome.

R8:

The four independent alleles of *osga3ox1* were confirmed by sequencing (Supplementary Figure 3b), and we analyzed the phenotypes of the T₁ generations of homozygous mutants (*osga3ox1-cr1/osga3ox1-cr1* and *osga3ox1-cr7/osga3ox1-cr7*), and confirmed that they showed the same phenotypes (Supplementary Figure 3e). In addition, the results of GA treatment and complementation experiments indicate that these phenotypes are caused by mutations in *OsGA3ox1* (Figures 2c, 5d and 5e). We mainly used the line with *osga3ox1-cr1/osga3ox1-cr1* for further phenotypic analyses.

According to the suggestion, we have added to the following sentence in line 190-192; “Similar phenotypes were observed using two independent mutant alleles, *osga3ox1-cr1* and *osga3ox1-cr7* (Supplementary Figure 3e). We mainly used *osga3ox1-cr1* for further analyses as *osga3ox1*.”

Supplementary Figure 3b and e

Q9:

-Ext Data Fig. 1a: Please, indicate the synthesis of GA3 and GA7 also in the scheme

R9:

We added GA₇ to the scheme in Supplementary Figure 1a. Since GA₃ is a GA produced by the

“Bakanae” fungus, and its synthetic pathway in rice is undetermined, only GA₇ was added. We have also added an explanation of GA₃ synthesis by “Bakanae” fungus in the Introduction section (line 72-78).

Scheme of gibberellin synthesis in rice (Supplementary Figure 1a)

Q10:

-Careful language editing is required: line 29 “had been occurred”; line 36 “is likely account”; line 204 “pettern”; line 299 “this experiments”; lines 336, 342 “straining” and other parts of the manuscript.

R10:

We carefully rewrote the manuscript, and we used an English proofreading service to check for language-related errors.

Reviewers' comments:

Reviewer #1 (Remarks to the Author):

In this version, the authors have done an extraordinary job at addressing all the queries previously raised. I appreciate the effort, especially under covid restrictions, to generate new experimental evidence. The result is very good and I think that the conclusions are conservative and in tune with the data presented.

Reviewer #2 (Remarks to the Author):

The authors addressed most of the questions raised in the previous round of review. Even with the new experiments, the findings are "consistent" with the proposed importance of GA7 for anther development, but a direct test of this importance is not provided. Similarly, the negative effect of producing GA7 in other parts of the plant is not directly tested. In consequence, the authors should be careful when expressing their conclusions about these points.

Regarding the western blot shown in Fig. 2k, the authors must provide a proof of equal protein loading when comparing OsCIN3 levels in Nipponbare vs. the *osga3ox1* mutant in each fraction.

Line 78: Do the authors refer to ent-kaurene?

We are very grateful to the two reviewers for once again reviewing our manuscript and providing valuable comments and suggestions. Listed below are our detailed point-by-point responses to the comments and issues raised by the reviewers. We hope that the following responses address all the concerns.

The areas that have been revised are highlighted in yellow in the main manuscript.

Our responses to the Reviewers' comments are as follow:

Responses (R) to editor and reviewer's comments/questions (Q)

RESPONSE TO REVIEWER #1:

In this version, the authors have done an extraordinary job at addressing all the queries previously raised. I appreciate the effort, especially under covid restrictions, to generate new experimental evidence. The result is very good and I think that the conclusions are conservative and in tune with the data presented.

We would like to thank you for your suggestions which helped improved our manuscript greatly.

RESPONSE TO REVIEWER #2:

The authors addressed most of the questions raised in the previous round of review. Even with the new experiments, the findings are "consistent" with the proposed importance of GA7 for anther development, but a direct test of this importance is not provided. Similarly, the negative effect of producing GA7 in other parts of the plant is not directly tested. In consequence, the authors should be careful when expressing their conclusions about these points.

We appreciate the concerns regarding the conclusions. We have revised the manuscript as indicated below:

Lines 127–128:

We believe that GA₇ may have the highest efficiency among bioactive GAs in promoting shoot elongation

Lines 340–342:

These results indicate that the ability to synthesize GA₇ as well as GA₄ may be important, and is likely related to pollen starch accumulation and improved reproductive performance.

Lines 351–353:

Given the male sterility phenotype of *osga3ox1*, during evolutionary processes, *GA3ox1* may have become crucial for male fertility in rice.

Q1:

*Regarding the western blot shown in Fig. 2k, the authors must provide a proof of equal protein loading when comparing OsCIN3 levels in Nipponbare vs. the *osga3ox1* mutant in each fraction.*

R1:

As suggested, we have added the image of the CBB staining of the total fraction to the western blot analysis depicted in Figure 2k to show the equal protein loading of the whole anther (Figure 2k and Supplementary Figure 13b). However, the images of fraction A–E are not shown because the amount and composition of protein of each fraction are different.

The following text was added to the legend of Figure 2k:

CBB staining of total extract is also shown. (Line 748)

The following text has been added to the legend of Supplementary Figure 13:

Supplementary Figure 13. Uncropped images of western blot and the CBB-stained gel. Full images of western blot (a) and CBB staining (b) that were depicted in part in Figure 2k. (Lines 1156–1158)

Q2:

Line 78: Do the authors refer to ent-kaurene?

R2:

Yes. We apologize for the oversight and we have corrected the typo. (Line 78)